# Livelihood Vulnerability Index: Gender Dimension to Climate Change and Variability in REDD + Piloted Sites, Cross River State, Nigeria

Adeniyi Okanlawon Basiru [1,*], Abiodun Olusegun Oladoye [1], Olubusayo Omotola Adekoya [2], Lucas Aderemi Akomolede [2], Vincent Onguso Oeba [3], Opeyemi Oluwaseun Awodutire [4], Fredrick Charity [5] and Emmanuel Kolawole Abodunrin [6]

[1] Department of Forestry and Wildlife Management, College of Environment Resources Management, Federal University of Agriculture, Abeokuta PMB 2240, Ogun State, Nigeria
[2] Forestry Research Institute of Nigeria, Ibadan PMB 5054, Oyo State, Nigeria
[3] Climate Change Department, Kenya Forestry Research Institute, Muguga off Nairobi-Nakuru Highway, Nairobi County P.O. Box 20412-00200, Kenya
[4] Department of Forestry Technology, Oyo State College of Agriculture, Igboora PMB 10, Oyo State, Nigeria
[5] Department of Forestry and Wildlife Management, Faculty of Agriculture, University of Port Harcourt, East/West Road, Choba PMB 5323, River State, Nigeria
[6] Department of Forestry Technology, Federal College of Forestry, Ibadan PMB 5054, Oyo State, Nigeria
[*] Correspondence: okanlawon.basiru@students.jkuat.ac.ke; Tel.: +234-813-833-4542

**Abstract:** Vulnerability to climate change and variability impacts has been identified as a major cog in the wheel of both livelihood and resilience, particularly in vulnerable groups in rural areas. This study aims to assess genders' vulnerability dimension to climate change and variability in REDD + (Reducing Emission from Deforestation and Forest Degradation+) piloted site/clusters, Cross River State, Nigeria. Data were proportionately collected from selected 200 respondents on gender disaggregated level using questionnaires. The assessment adopted the sustainable livelihood approach (livelihood vulnerability index) and compared the results with the IPCC vulnerability standard of exposure, sensitivity and adaptive capacity weighted mean. The results revealed a significant difference in the vulnerability dimension of both women and men disaggregated levels (LVI: men 0.509, women 0.618). The women category was more vulnerable to six out of seven major components of LVI assessed: (livelihood strategies (0.646), social networks (0.364), water (0.559), health (0.379), food and nutrition (0.507), and natural hazards and climate variability (0.482), while men only vulnerable to socio-demographic major component (0.346). Vulnerability indices also showed women to be more exposed (0.482), and sensitive (0.489) with the least adaptive capacities (0.462) to the climate change and variability impacts. Overall, on the IPCC-LVI index, women are more vulnerable (0.0098) to climate change and variability impacts than men (−0.0093). The study recommends that the women's category resilience and adaptive capacity should be empowered in adaptation projects in climate change such as REDD + (Reducing Emissions from Deforestation and Forest Degradation+) to reduce their vulnerability to impacts of climate change and variability in the context of exposure, sensitivity, and adaptive capacities. This will be instrumental in formulating policies to address the specific needs of gender categories in reducing vulnerability to climate change and variability. This pragmatic approach may be used to monitor gender vulnerability dimension, and livelihood enhancement and evaluate potential climate change adaptation programs. Additionally, the introduction of IPCC-LVI as a baseline instrument will enhance information on gender resilience and adaptive capacity for policy effectiveness in a data-scarce region particularly Africa.

**Keywords:** livelihood; exposure; sensitivity; adaptive; capacity; index

## 1. Introduction

Climate change and climate variability are global phenomena that have caused serious concern to many sectors of the economy and livelihoods, predominantly rural dwellers. According to Intergovernmental Panel on Climate Change (IPCC) [1], climate change refers to "a change in the state of the climate that can be identified (using statistical tests) by changes in the mean and/or the variability of its properties, and that persists for an extended period, typically decades, or longer. Communities all over the world encounter changes and events that impact them both positively and negatively in their lives. Africa is presumed to be more vulnerable to the risk of climate change compared to other continents" [2]. Africa is prone to the impacts of climate change because of the reliance on natural resources, non-irrigated agriculture, and limited adaptive capacity [3]. Climate change and variability affected gender and access to natural resources differently based on the different capacity/roles. [4]. Therefore, the effect of climate change and variability is expected to differ based on agro-ecological regions, spatial features, and socio-economic groups such as gender [5]. "Vulnerability is a function of the character, magnitude, and rate of climate change and variation to which a system is exposed, its sensitivity, and its adaptive capacity" [6] while UNDRR [7] defines vulnerability as "the conditions determined by physical, social, economic and environmental factors or processes which increase the susceptibility of an individual, a community, assets or systems to the impacts of hazards". The two definitions were synthesized by Hertel and Rosch, [8] describing vulnerability to the impact of climate change as dynamic, locally specific, and manifested along gender, social, and poverty lines. Therefore, understanding gender equity and their connection are important in decision-making on climate change adaptation and mitigation discussion. This is crucial because strategies to address gender plight concerning climate change are unclear or vague [9]. Although it has been recorded that most studies on climate change including millennium development goals (MDGs) and sustainable development nexus lack a gender-focused analysis [10,11], these studies on climate change have often focused on poverty eradication, while there is silence on the gender–vulnerability linkage dimension to climate change and variability considering the sustainable livelihood approach in adaptation and mitigation action plan such as REDD + (Reducing Emission from Deforestation and Forest Degradation+).

Different studies have suggested and recommended that decision-making and approaches that involve gender in combating climate change mitigation and adaptation from the grassroots to the national level will go a long way in addressing gender inequality and vulnerability problems [12–15]. Available studies conducted where gender and climate change vulnerability are captured include the study of [16–20], among others. None of these studies addressed gender and climate change in relation to social role and constructed responsibility with vulnerability dimension using SLA (sustainable livelihood approach) in the context of IPCC-LVI index using capitals frameworks (natural, physical, financial, human, and social). On the other hand, the majority of the empirical studies that assessed gender vulnerability to the effect of climate change focused on farm decision-making related to adaptation, perception and adaptation, variation in farm household vulnerability, and measurement of climate vulnerability across ecological zones. Therefore, despite the importance and potential of gender in mitigating and adapting to the effect of climate change and climate variability, no study has been carried out to bring out the efficacy of social and cross-sectional roles to determine their vulnerability dimensions. The information is important in enabling the vulnerable/social groups to develop adaptive capacity/measures that will shed more light on the dimension of vulnerability and coping mechanisms. The outcomes should also be fed into the climate negotiating process to enable decision makers to have a better understanding and in-depth of how different groups of people is affected and what adaptive capacity and support is needed in holistic approaches to tackle the menace of climate change impact. Therefore, the objective of this study is to examine the vulnerability dimension of men and women categories to climate change and variability in REDD + (Reducing Emissions from Deforestation and Forest Degradation+) piloted

site/clusters, Cross River State, Nigeria. The authors suggest that there is a significant difference in the vulnerability dimension of the men and women categories. Apart from adding to the body of knowledge, the findings of this study will provide definite gender vulnerability levels, sensitivity, and adaptive capacities to climate change and variability. This will be instrumental in formulating policies to address the specific needs of gender categories in reducing vulnerability to climate change and variability as a way of achieving Nigeria REDD + and National Environmental, Economic and Development Study (NEEDS) goal statements objective of gender mainstreaming and equity.

## 2. Materials and Methods

### 2.1. Study Area

2.1.1. Location

Cross River State (CRS) is one of the 6 states that are located around the coast of Niger Delta in the southern part of the country. Geographically, Cross River State is located between latitude 4° 28′ and 6° 55′ North of the equator and, Longitude 7° 50′ and 9° 28′ East of the Greenwich Meridian. It shares the same boundaries with Benue State in the north, Atlantic Ocean in the South, Abia and Ebonyi states in the West, and an extensive border with the Republic of Cameroon in the East. For this study, three key sites (known as REDD + piloted sites/clusters) were purposively selected as the only approved piloted sites/clusters for the on-going United Nation's REDD + program (Reducing Emissions from Deforestation and Forest Degradation+) currently ongoing in CRS, Nigeria. The sites/clusters of interest include: The Afi–Mbe, Ekuri–Iko, and mangrove forest clusters (about 16 communities with approximately a population size of 30,000 peoples) from which communities were selected accordingly (Figure 1). The Afi–Mbe REDD + site/clusters border the Cross River State Forest Reserve. The reserve lies between the Afi Mountain Wildlife Sanctuary and Mbe Mountains Community Forest while Ekuri–Ukpon clusters are made up of community forests and forest reserves, jointly managed by local communities, the government (Cross River Forestry Commission) and the conservation Society). In the Afi–Mbe cluster, the existing protected areas include the Afi Mountain Wildlife Sanctuary, Afi River Forest Reserve, Mbe Mountains and a community forest south of the Cross River National Park. The Ekuri–Iko cluster is made of the Ukpon River Forest Reserve, Ekuri Community Forest, parts of the Oban Block Forest Reserve and the Cross River South Forest River Ekuri Cluster, a collection of communities located on the edge of the Cross River National Park buffer zone, while the mangrove forest was bordered by the creek in Akpabuyo local government of the state [21].

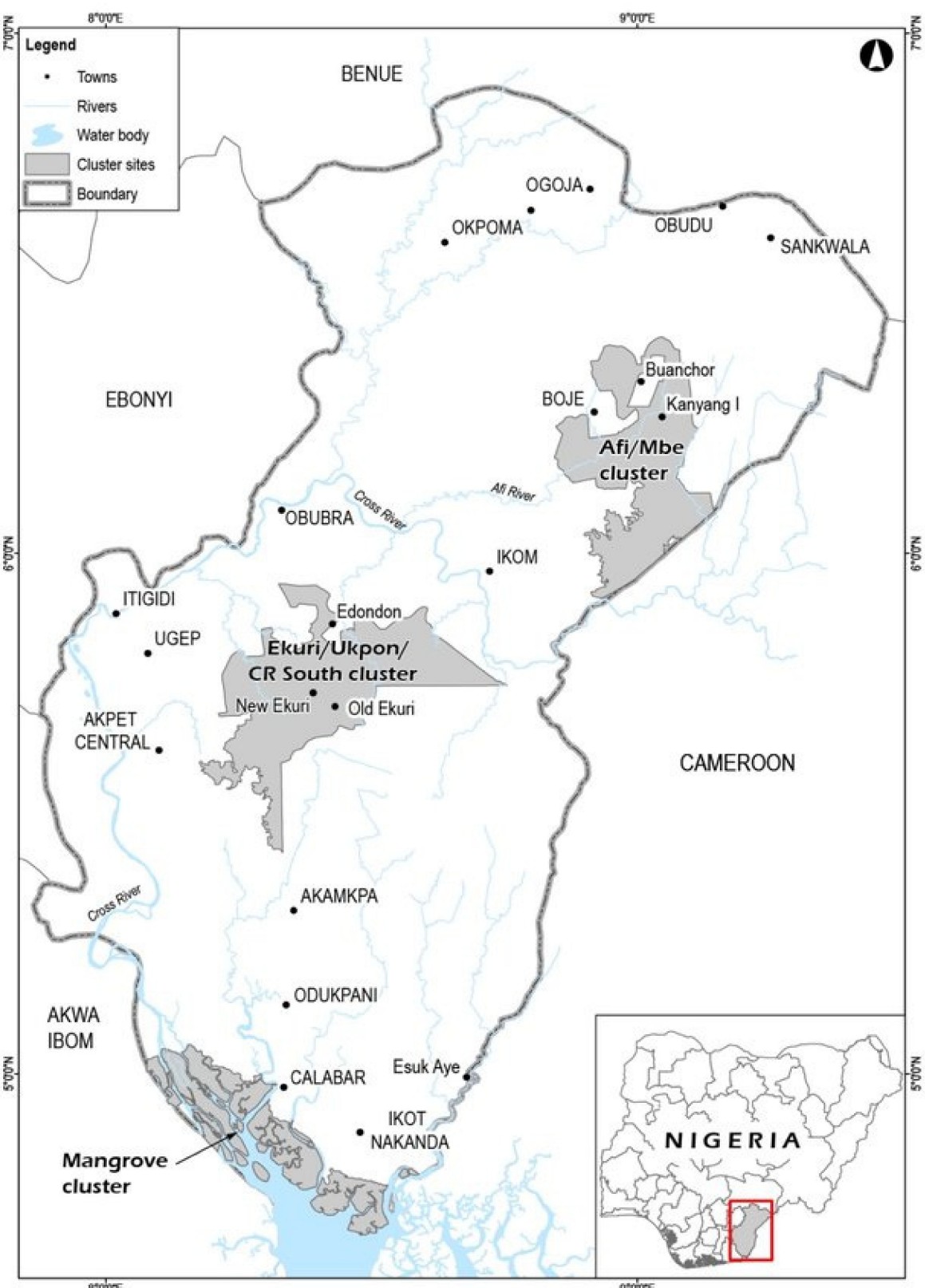

**Figure 1.** Map of Cross River State showing the three cluster sites with Insert Map of Nigeria. Adapted from Onojeghuo et al. 2016 [22].

Relief and Vegetation

The relief of Cross River State consists of the coastal creeks towards the southern border with the Atlantic Ocean, Cameroon Mountains, and part of Bamenda highland in the east, as well as the Cross River basin in the west. Altitude ranges from sea level, gently undulating basins to volcanic hills of Oban and Ogoja that extend up to 6000 feet.

Cross River State has 4 main types of vegetation that reflect the main ecological zones within the state. These are (1) freshwater swamps and mangroves (2) evergreen wet forests (3) southern guinea savanna, and (4) montane forests and grasslands.

These vegetation zonations are greatly influenced by the topography of the area. The mangrove belt covers about 10–15 km along the coast where the ocean mixes with fresh waters. Predominantly, the mangrove trees are shrubby with heights of about 40 m, consisting of both local and exotic species of palm trees and Rhizophora. The freshwater swamp has a wider coverage of about 10–25 km extending towards the north of the mangrove belt. The height of the freshwater swamp forest canopy is about 30 m and consists of mostly woody and non-woody species arranged in different layers. The largest portion of forests in the state is the evergreen lowland rainforest which extends southwest into Cameroon. This zone is considered the remaining pristine rainforest vegetation in the whole of Nigeria and has been managed by Cross River National Park, forest reserves, and indigenous forest communities [23].

In Cross River State, savannah-like vegetation is found around the northern and central portions consisting of various species of trees and grasses. Montane vegetation is also seen around the north-eastern portion on the border with Cameroon. These areas include the Obudu Plateau, Sankwala Mountains, and Ikwete hills with elevations of about 1800 m above sea level. This place is of high species richness and diversity including both vascular and nonvascular plants that reflect the local microclimatic conditions [23].

Climate

In Cross River, there is an annual alternation of distinct wet and dry seasons mostly determined by the movement of Inter Tropical Convergence Zone (ITCZ). Normally, annual rainfall starts in April and ends in October with a peak usually in August in most parts of the country. However, the southern regions experience 4 distinct seasons consisting of (1) a long rainy season from February to July, (2) a period of decline known as August break, (3) a short period of heavy rainfall from September to November (4) dry season from mid-November to February. The amount of rainfall decreases northwards from the coastal regions with an annual range of 1854 mm and 508 mm, respectively.

In some remote corners of the north-east, especially near the border with Chad, annual rainfall can be as low as 1 inch for 5–7 months. There is also temperature variability throughout the country. Annual mean maximum temperatures could be up to 36 degrees centigrade in the northern savannah regions, while the annual mean minimum temperature of 23° Fahrenheit is usually recorded in the southern regions. The mean annual temperature in Cross River State ranges from 22.4 °C to 30.1 °C. Additionally, mean annual rainfall also varies significantly locally from 2018 mm to 3063 mm [24].

2.1.2. Study Design and Population

Nigeria applied for membership of the (United Nation's Reducing Emissions from Deforestation and Forest Degradation+) UN-REDD + in December 2009 [23] and it's REDD + readiness plan was approved for funding in October 2011 [25]. Nigeria-REDD + (Reducing Emissions from Deforestation and Forest Degradation+) has a national program and a state-level program with Cross River State as the pilot site. At the national level, the Nigeria-REDD + Secretariat is housed in the Department of Climate Change at the ministry for the environment. This ministry works closely with the national advisory council on Reducing Emissions from Deforestation and Forest Degradation+ (REDD +) and the national technical REDD + committee. The advisory council is a policy-making body, while the technical committee is a working group comprising of UN-REDD + and Nigeria-REDD + (national

and state level) personnel. In addition, at the national level, there is the REDD + steering committee, which is another working group for effective coordination of the work of the Department of Climate Change and the Cross River State Forestry Commission [25]. There is also a national civil society organizations' REDD + forum, a platform for civil society to have a voice in Nigeria-REDD + through the Department of Climate Change.

The recommendations for the finalization of the program document by all the stakeholders in design and implementation of the Nigeria REDD + readiness program are as follows:

(i.) Forest communities should be properly engaged, receive training, and feel early and tangible actions throughout the program's implementation;

(ii.) here is need for REDD + to have a broad approach that goes beyond forest conservation to address questions of land management, afforestation and reforestation, ecosystem restoration, sustainable agriculture and community-based livelihoods;

(iii.) There is need for capacity building on forest monitoring systems;

(iv.) The program should include provisions to assess issues of land tenure, carbon rights, fair benefit-sharing mechanisms, and community conflict, providing guidance on how to address them in the context of REDD +.

According to design and implementation of the REDD + readiness program signed, few components were highlighted with emphasis and attention focused more on issues of consultation, forest governance, community rights, *enhancement of sustainable livelihoods, welfarism and gender equality* [26].

Quantitative approaches were adopted for this study. The population for this research study consists of some selected communities (from clusters) where climate change adaptation initiatives such as REDD + is being piloted was purposefully selected based on UN-REDD + on-going project recommended site [27].

### 2.1.3. Sampling Procedure and Methodology

Cochran [28], formulae was used to determine the right population proportion for this study. This was projected to be:

$$n = \frac{N}{(1 + N(e))^2} \tag{1}$$

where:

$n$ = Sample size
$N$ = Total population
$e$ = Desired level of precision.

A multi-stage sampling technique was used. The first stage involved a purposive selection of Afi–Mbe, Ekuri–Iko, and mangrove site/clusters, as these are the approved Reducing Emissions from Deforestation and Forest Degradation+ (REDD +) piloted site/clusters in Nigeria. The second stage adopted the proportion sampling method to select two communities each from Afi–Mbe, Ekuri–Iko and mangrove site/clusters. The number of households selected from each community was also based on the proportion sampling technique. Within each community selected, all the gender in the households were listed and stratified into men and women along age brackets (youth: 18–35 years, men/women: 35–60 years, and elderly: 60 years and above) [29], and then, simple random sampling was used to select the required number of men and women to constitute the sample units to whom questionnaire was later administered (Figure 2). In all, 200 gendered-disaggregated respondents were interviewed proportionately, 100 men and 100 women.

### 2.1.4. Data Collection

Instrument of Data Collection

The research study employed questionnaires as tools/instruments for data collection. The main strength of quantitative measurement instrument in this study is that the researcher has control over the topics and the format of the interview. Moreover, this tool was

utilized based on its established nature, prominence, popularity/acceptance, adaptability, and the potential it offered in helping to obtain the data required for this type of study.

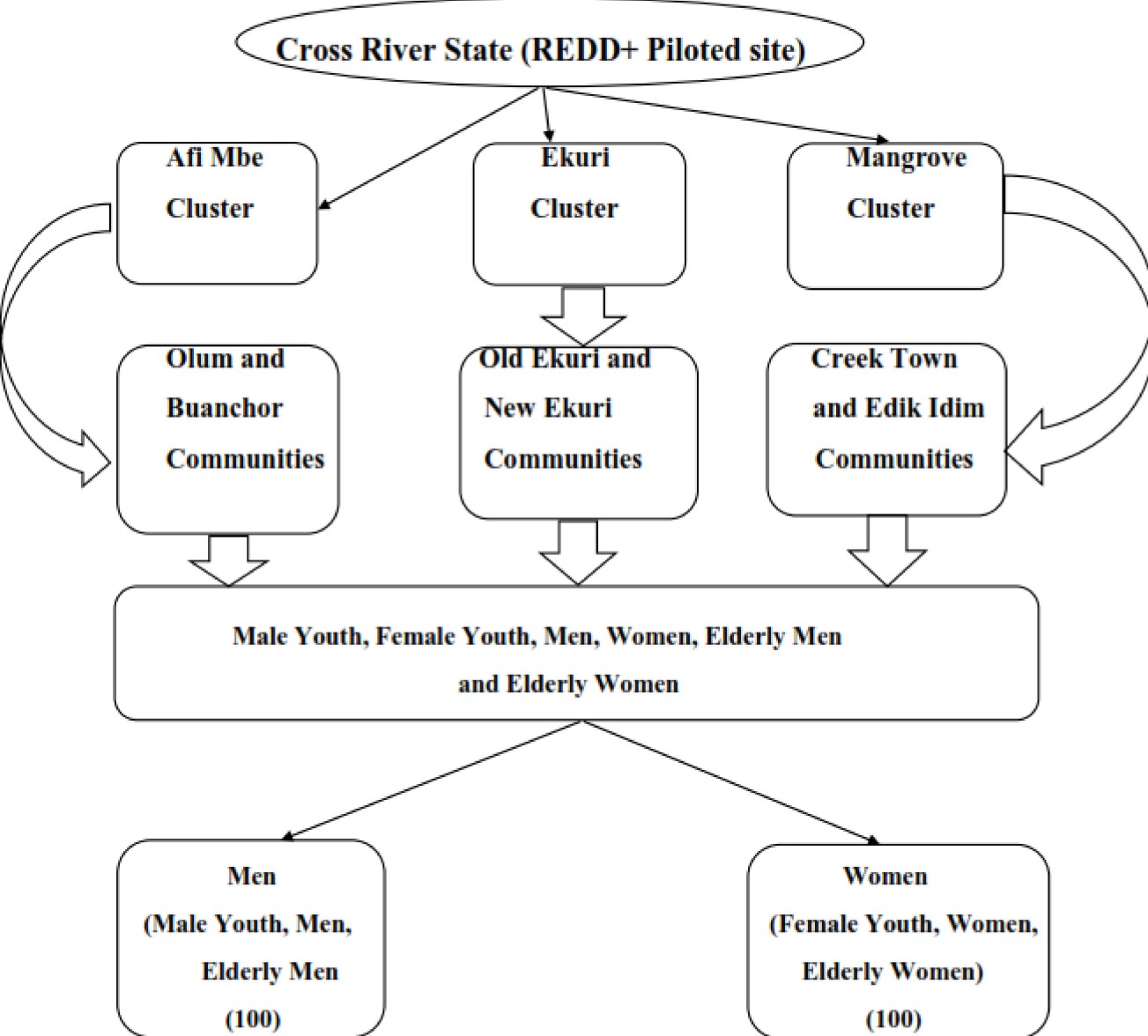

**Figure 2.** Sampling workflow diagram for the study (source: authors).

A total number of 200 questionnaires were administered between June 2021 to December 2021 by trained professional forest officers, an instructor from the Cross River State Ministry of Forestry, and research assistant personnel from the Department of Forestry and Wildlife, Federal University of Agriculture, Abeokuta, Nigeria. The method of administration was performed by selecting the respondents from the households in the community until the required respondents are selected to avoid possible bias and achieve actual representation of the selected communities.

The questionnaire contains both structured questions with rated response answers and open-ended questions requiring short answers that were developed through an informal survey to the study site before an effective formal survey. Rather than handing the questionnaires to the respondents to fill themselves, interviewers were used to filling the forms for better collection of data given that the level of education is expected to be low.

Validation and Reliability of Multi-Item Measures

Construction of instrument validity is necessary as recommended by the different literature on methods of using questionnaires to collect data. This was performed after careful selection of contents of the interview schedule and passing through a series of critical examinations to certify its content and face validities. The instrument was subjected to critical scrutiny and consequent modification by other researchers to guarantee its content validity.

Pre-Testing of Data Collection Tools

Pre-testing of the structured interview schedule was performed, and the instrument was implemented to collect the data required for this study. The data were collected by a team of professional forest officers and data collection personnel from the Cross River State Ministry of Forestry. The structured and open-ended questions were designed in a way to make data management and analysis easy through numerical coding of responses. After coding the various responses, data were entered into MS Excel, removing non-meaningful responses, and then analyzed with SPSS version 25 using descriptive and inferential statistics and a vulnerability radar diagram.

IPCC Livelihood Vulnerability Index

The sustainable livelihoods approach [30], which considers five categories of household assets: social, financial, physical, human, and natural capital is a method for developing community development initiatives [31]. The method has shown to be beneficial in determining a household's ability to resist shocks such as epidemics or civil conflict. Climate change complicates the security of household livelihoods. The sustainable livelihoods approach addresses issues of sensitivity and adaptive capacity to climate change to a limited extent, but in order to comprehensively evaluate livelihood risks resulting from climate change, a new vulnerability assessment approach that integrates climate exposures and accounts for household adaptation practices is required. To evaluate the differential impacts of climate change on populations in six UN-REDD + piloted communities in Cross River State, Nigeria, existing methodologies to create a new livelihood vulnerability index (LVI) were performed. The LVI assesses gender category vulnerability to natural hazards and climate variability, as well as their present health, food, and water resource features, which determine their sensitivity to climate change impacts. The LVI is expressed in two ways: the first as a composite index made up of seven key components, while the second combines the seven into the IPCC's three contributing variables to vulnerability: exposure, sensitivity, and adaptive capability.

The method used in this study differs from previous methods because it constructed the indexes using primary data from households with different gender categories along the age line. It also included a framework for combining and aggregating indicators at the gender disaggregated level, which is useful for planning development and adaptation. This strategy circumvents the difficulties associated with secondary data by employing primary household data. Another benefit is that it reduces reliance on climate models, which, despite recent improvements, are still presented at too big a scale to produce reliable estimates at levels suitable for community development planning [32,33]. Owing to a lack of climatic data, human resources, and computer equipment, regional climate forecasts are likely to mask inequalities in vulnerability across populaces within nations such as Nigeria, which have terrain ranging from lowland coastal plains to mountains, as well as variable degrees of infrastructure and socio-economic development. Rather than focusing on climate forecasts, the LVI method focuses on evaluating the robustness of present livelihood and health systems, as well as communities' capability to change these strategies in response to climate-related exposures.

The LVI is a useful instrument for development organizations, policymakers, and public health practitioners to identify the demographic, socioeconomic, and health elements that contribute to climate vulnerability at the household, district, and community levels.

It is built to be adaptable, allowing development planners to fine-tune and focus their studies to meet the specific demands of each geographic location. Sectoral vulnerability ratings can be separated from the overall composite index to identify relevant intervention locations. Vulnerability to climate change is intimately linked to gender when it is viewed as a condition of well-being that differs among people based on their resource endowments and social hierarchical location [34]. However, there has been no definitive study on the specific relationship between gender and vulnerability. In the context of this study, according to IPCC, vulnerability is the extent to which gender disaggregated levels in the household is susceptible to, or unable to adapt to, the negative effects of climatic stresses.

Calculating the LVI: Composite Index Approach

The livelihood vulnerability index (LVI) uses a balanced weighted average approach [35] where each sub-component contributes equally to the overall index even though each major component is comprised of a different number of sub-components. Because this study intends to develop an assessment tool accessible to a diverse set of users in resource-poor settings, the LVI formula uses the simple approach of applying equal weights to all major components. This weighting scheme could be adjusted and adopted by future users as needed.

Because each of the sub-components is measured on a different scale, it was first necessary to standardize each as an index. The equation used for this conversion was adapted from that used in the human development index to calculate the life expectancy index, which is the ratio of the difference of the actual life expectancy and a pre-selected minimum, and the range of predetermined maximum and minimum life expectancy [36]:

$$Index_{SG} = \frac{S_G - S_{min}}{S_{max} - S_{min}} \tag{2}$$

where $S_G$ is the original sub-component for gender $g$, and $s$ and $s$ are the minimum and maximum values, respectively, for each sub-component determined using data from both gender categories. For example, the "average time to travel to primary water source" sub-component ranged from 1 to 300 min among the two gender categories surveyed. These minimum and maximum values were used to transform this indicator into a standardized index so it could be integrated into the water component of the LVI. For variables that measure frequencies such as the "percent of genders reporting having heard about conflicts over water resources in their community," the minimum value was set at 0 and the maximum at 100. Some sub-components such as the average non-timber forest products (NTFP) livelihood diversity index were created because an increase in the crude indicator, in this case, the number of livelihood activities undertaken by gender, was assumed to decrease vulnerability. In other words, this study assumed that an individual who farms and raises animals is less vulnerable than a category who only farms. By taking the inverse of the crude indicator, the creation of numbers was performed to assign higher values to gender with a lower number of livelihood activities. The maximum and minimum values were also transformed following this logic and Equation (2) was used to standardize these sub-components.

After each was standardized, the sub-components were averaged using Equation (3) to calculate the value of each major component:

$$M_G = \frac{\sum_{i=1}^{n} indexS_G}{n} \tag{3}$$

where $M_G$ = one of the seven major components for gender (g) (socio-demographic profile (SDP), livelihood strategies (LS), social networks (SN), health (H), food and nutrition (FN), water (W), or natural hazards and climate variability (NHCV)), the index represents the sub-components, indexed by $i$, that make up each major component, and n is the number of sub-components in each major component. Once values for each of the seven major

components for the gender were calculated, they were averaged using Equation (4) to obtain gender dimension of LVI:

$$\text{LVI}_{\text{G}} = \frac{\sum_{i=1}^{7} WMiMGi}{\sum_{i=1}^{7} WMi} \tag{4}$$

which can also be express as

$$\text{LVI}_{\text{G}} = \frac{W_{SDP}\,SDP_G + W_{LS}\,LS_G + W_{SN}\,SN_G + W_H\,H_G + W_{FN}\,FN_G + W_W\,W_G + W_{NHCV}\,NHCV_G}{W_{SDP} + W_{LS} + W_{SN} + W_H + W_{FN} + W_W + W_{NHCV}} \tag{5}$$

where $\text{LVI}_G$, the livelihood vulnerability index for gender $g$, equals the weighted average of the seven major components. The weights of each major component, $WMi$, are determined by the number of sub-components that make up each major component and are included to ensure that all sub-components contribute equally to the overall LVI. In this study, the LVI is scaled from 0 (least vulnerable) to 1 (most vulnerable) [37].

The livelihood vulnerability index (LVI) includes seven major components: socio-demographic profile, livelihood strategies, social networks, health, food and nutrition, water, and natural hazards and climate variability. Each is comprised of several indicators or sub-components (Table 1). These were developed based on a review of the literature on each major component, as well as the practicality of collecting the needed data through household surveys. Table 1 and Appendix A includes an explanation of how each sub-component was quantified, the survey question used to collect the data, the original source of the survey question, and potential sources of bias.

**Table 1.** Major components and sub-components comprising the livelihood vulnerability index (LVI) developed for gender in REDD + piloted site/clusters in Cross River State, Nigeria.

| Major Components | Sub-Components | Explanation of Sub-Components | Source |
|---|---|---|---|
| Socio-demographic profile | Dependency ratio | Ratio of the population under 15 and over 19 and 64 years of age to the population between 19 and 64 years of age | Adapted from Hahn et al. [38] |
| | Average gender age of female household head | Average age of household that happens to be female | Purposefully developed for this questionnaire |
| | % of female-headed households | Percentage of household where the primary adult is women. If a male-head is away from the home > 6 months per year the female is counted as the head of the household | Adapted from Hahn et al. [38] |
| | % of gender where the head of household have formal education | Percentage of genders where the head of the household reports that they have attended formal school. | Adapted from Hahn et al. [38] |
| | % of gender recorded orphans less than 13 years of age | Percentage of gender that have at least 1 orphan living in their home. Orphans are children < 13 years old who have lost one or both parents. | Adapted from Hahn et al. [38] |
| Livelihood strategies | % of gender depend solely on forest resources as source of income | Percentages of gender whose gathering of forest resources determines their source of income | Purposefully developed for this questionnaire |
| | % of households with family member working in a city or foreign country | Percentage of genders that report at least 1 family member who works outside of the community for their primary work activity | Adapted from Hahn et al. [38] |
| | % of gender raising animals | Percentage of gender that report raising animals for livelihood enhancement | Purposefully developed for this questionnaire |

**Table 1.** *Cont.*

| Major Components | Sub-Components | Explanation of Sub-Components | Source |
|---|---|---|---|
| Health | % of gender growing crops | Percentage of gender that report growing crops for livelihood enhancement | Purposefully developed for this questionnaire |
| | % of gender collect forest resources from bush, forest or water | Percentage of gender that reported collecting forest resources for income and consumption | Purposefully developed for this questionnaire |
| | Livelihood diversification index (range: 0.25–1) [a] | The inverse of (the number of agricultural and NTFPs livelihood activities +1) reported by a genders, e.g., an individual that farms, raises animals, and collects natural resources will have a livelihood diversification index = 1/(3 + 1) = 0.25. | Adapted from Hahn et al. [38] |
| | Average time to health facility (minutes) | Average time it takes the genders to get to the nearest health facility. | Adapted from Hahn et al. [38] |
| | % of gender with family member with chronic illness | Percentage of gender that report at least 1 family member with chronic illness chronically ill (they get sick very well, chronic illness was defined subjectively by often) | Adapted from Hahn et al. [38] |
| | % of gender where a family member had to miss work or school in the last 3 weeks due to illness | Percentage of gender that report at least 1 family member who had to miss school of work due to illness in the last 3 weeks | Adapted from Hahn et al. [38] |
| | % of gender recorded death due to climate-related disaster | Percentage of family member climate-related disaster has claimed their lives | Adapted from Hahn et al. [38] |
| | Average malaria exposure prevention index (range: 0–12) | Months reported exposure to malaria Owning at least one bednet indicator | Adapted from Hahn et al. [38] |
| | % of gender that has mosquito net/bednet | Percentage of gender with ownership of net/bednet | Adapted from Hahn et al. [38] |
| Social Networks | Average Receive: Give ratio (range: 0–15) | Ratio of (the number of types of help received by gender in the past month + 1) to (the number of types of help given by gender to someone else in the past month + 1). | Adapted from Hahn et al. [38] |
| | Average borrow: lend money ratio (range: 0.5–2) | Ratio of gender borrowing money in the past month to gender lending money in the past month, e.g., if gender borrowed money but did not lend money, the ratio = 2:1 or 2 and if they lend money but did not borrow any, the ratio = 1:2 or 0.5. | Adapted from Hahn et al. [38] |
| | % of gender that their family have not gone to their local government for assistance in the past 12 months | Percentage of gender that reported that they have not asked their local government for any assistance in the past 12 months | Adapted from Hahn et al. [38] |
| | % of gender belongs to NGOs/affiliation society | Percentage of NGOs/affiliation society the gender category belongs to in the community | Purposefully developed for this questionnaire |
| Food and Nutrition | % of gender that get sufficient food for the whole year | Percentage of gender that experience availability of food throughout the year | Purposefully developed for this questionnaire |
| | Average number of months gender struggle to find food (range: 0–12) | Average number of month's gender struggle to obtain food for themselves. | Adapted from Hahn et al. [38] |

<div style="text-align:center">**Table 1.** *Cont.*</div>

| Major Components | Sub-Components | Explanation of Sub-Components | Source |
|---|---|---|---|
| | % of gender depend solely on non-timber forest products (NTFPs) | Percentage of gender that depend solely on NTFPS gathering for consumption and income | Purposefully developed for this questionnaire |
| | Average crop diversity index (range: >0–1) [a] | The inverse of (the number of crops grown by gender +1). e.g., an individual that grows pumpkin, maize, okra beans, and cassava will have a crop diversity index = 1/(4 + 1) = 0.20. | Adapted from Hahn et al. [38] |
| | Average NTFP diversity index (range: >0–1) [a] | The inverse of (the number of NTFP by gender +1), e.g., an individual that collected leaves, snails, rattan, mushroom, and fruits will have a NTFP diversity index = 1/(4 + 1) = 0.20. | Purposefully developed for this questionnaire |
| | % of gender that do not save crops | Percentage of gender that do not save crops from each harvest. | Purposefully developed for this questionnaire |
| | % of gender that do not save NTFPs | Percentage of gender that do not save NTFPs resources from year to year. | Purposefully developed for this questionnaire |
| | % of gender that suffers from any kind of nutritional deficiency | Percentage of gender that suffers any kind of nutritional deficiency | Purposefully developed for this questionnaire |
| | % of gender source of energy for cooking especially firewood | Percentage of gender uses firewood as source of energy for cooking | Purposefully developed for this questionnaire |
| | % of gender collecting fuelwood for cooking | Percentage of gender collecting fuelwood for cooking | Purposefully developed for this questionnaire |
| | Average time gender spent to collect firewood/fuelwood from the forest | Average time an individual spent in collecting fuelwood/firewood from the forest | Purposefully developed for this questionnaire |
| | % of gender ascertain firewood availability has reduced in the last 10 years | Percentage of gender perception of fuelwood availability over 10 years | Purposefully developed for this questionnaire |
| | % of gender using traditional methods of cooking | Percentage of gender using traditional methods for cooking | Purposefully developed for this questionnaire |
| Water | % of gender that do not have a constant water supply | Percentage of gender that suffers inconsistent water supply in the community | Purposefully developed for this questionnaire |
| | % of gender that do not have a clean or safe water | Percentage of gender that doesn't have access to clean or safe water in the community | Purposefully developed for this questionnaire |
| | % of gender reporting water conflicts | Percentage of gender that report having heard about conflicts over water in their community | Purposefully developed for this questionnaire |
| | Average time to water source (minutes) | Average time it takes gender to travel to their primary water source. | Adapted from Hahn et al. [38] |
| | % of gender that utilize a natural water source | Percentage of gender that report a creek, river, lake, pool, or hole as their primary water source. | Adapted from Hahn et al. [38] |
| | Inverse of the average number of liters of water stored by gender category | The inverse of (the average number of liters of water stored by each gender + 1). | Purposefully developed for this questionnaire |
| Natural hazards and climate variability | % of climate change and variability occurrence in the study communities as reported by gender category | Percentage of time climate change have affected the study communities | Purposefully developed for this questionnaire |
| | % of gender that received a warning about the pending natural hazards | Did you receive a warning about the flood/erosion/drought before it happened? | Purposefully developed for this questionnaire |

**Table 1.** *Cont.*

| Major Components | Sub-Components | Explanation of Sub-Components | Source |
|---|---|---|---|
| | % reporting death of person or family member | Has anyone of your family died of any climate-related hazards? | Purposefully developed for this questionnaire |
| | % reported injuries during extreme event | Have anyone of your family member injured during extreme weather event? | Purposefully developed for this questionnaire |
| | % reported erratic rainfall pattern | Have you been experiencing erratic rainfall pattern in this area for the last 10 years | Purposefully developed for this questionnaire |
| | % of gender that reported reduction in NTFPs resources due to climate variability | What is the status of NTFPs resource in accordance with climate variability for the past 10 years in this community? | Purposefully developed for this questionnaire |
| | % reported high temperature | Has the level of temperature increases in this community over last 10 years? | Purposefully developed for this questionnaire |
| | % reported destruction of farmland and properties by erosion | Has erosion destroyed your farmland or properties before? | Purposefully developed for this questionnaire |

[a] Some indicators such as the Livelihood Diversity index were created because of an increase in the crude indicators i.e., the number of livelihood activities undertaken by gender decreases vulnerability. As a result, inverse of these were taken by creating numbers that reflects lines of reasoning, thereby assigning higher values to gender with a low number of livelihood activities.

Calculating the LVI–IPCC: IPCC Framework Approach

This study developed an alternative method for calculating the LVI that incorporates the IPCC vulnerability definition. Table 2 shows the organization of the seven major components in the LVI–IPCC framework. Exposure of the study population is measured by the number of natural hazards and climate variability exposure factors such as unstable rainfall patterns, floods, erosion, etc., that occur at the household level because their effects can be felt individually. For this study, exposure was indicated by the actual impacts of climate change on individuals in the household by identifying gender disaggregated groups that suffered the most impact such as the death of persons or livestock, damage to farm structures, and those that were forced into temporary migration. One caveat for this approach is that exposure is underestimated in the model as only those households that reported extreme cases of impact such as death, destruction of property, and temporary migration are shown to be exposed to climate change [38].

**Table 2.** Indexed sub-components, major components, and *t*-test statistic for gender in REDD + sites, Cross River State, Nigeria.

| Major Components | Sub-Components | Index of Sub-Components | | Major Component Indices | | Two Sample *t*-Test | |
|---|---|---|---|---|---|---|---|
| | | Men | Women | Men | Women | *t*-Value | *p*-Value |
| Socio-demographic profile | Dependency ratio | 0.047 | 0.056 | 0.346 | 0.320 | 4.458 | 0.000 |
| | Female-headed household | 0.089 | 0.235 | | | | |
| | Average age of female head-household | 0.714 | 0.071 | | | | |
| | Gender with households where head has attended formal school | 0.871 | 0.768 | | | | |
| | Gender with household orphans | 0.010 | 0.469 | | | | |
| Livelihood strategies | Gender with family member working in the city or foreign country | 0.679 | 0.50 | 0.540 | 0.646 | 3.706 | 0.000 |
| | Gender raising animals | 0.634 | 0.584 | | | | |
| | Gender that grows crops | 0.790 | 0.762 | | | | |

**Table 2.** *Cont.*

| Major Components | Sub-Components | Index of Sub-Components | | Major Component Indices | | Two Sample *t*-Test | |
|---|---|---|---|---|---|---|---|
| | | Men | Women | Men | Women | *t*-Value | *p*-Value |
| Social networks | Gender dependent solely on forest resources as a source of income | 0.28 | 0.531 | 0.121 | 0.364 | 2.976 | 0.005 |
| | Gender collect forest resources from bush, forest, or water | 0.795 | 0.876 | | | | |
| | Average agricultural and NTFP livelihood diversification index | 0.063 | 0.625 | | | | |
| | Average receive: give ratio | 0.053 | 0.156 | | | | |
| | Average borrow: lend money | 0.017 | 0.850 | | | | |
| | Gender that has not gone to the government for assistance | 0.150 | 0.33 | | | | |
| | Gender affiliated society or NGO | 0.265 | 0.119 | | | | |
| Health | Average time to health facilities | 0.199 | 0.176 | 0.331 | 0.379 | 2.521 | 0.008 |
| | Gender with family member down with chronic illness | 0.091 | 0.355 | | | | |
| | Gender where a family member had to miss work or school in the past 3 weeks due to illness | 0.111 | 0.341 | | | | |
| | Gender with family member died due to climate-related hazards | 0.051 | 0.070 | | | | |
| | Average malaria exposure * prevention index | 0.681 | 0.650 | | | | |
| | Gender that has mosquito net | 0.850 | 0.687 | | | | |
| Food and nutrition | Gender that solely depend on NTFP | 0.28 | 0.531 | 0.465 | 0.507 | 12.467 | 0.000 |
| | Average number of months gender having trouble in getting enough food | 0.611 | 0.598 | | | | |
| | Average crop diversity index | 0.043 | 0.040 | | | | |
| | Average NTFP Diversity Index | 0.046 | 0.051 | | | | |
| | Gender that does not save some crops | 0.574 | 0.670 | | | | |
| | Gender that does not save some NTFP resources | 0.554 | 0.618 | | | | |
| | Gender that gets sufficient food for the whole year | 0.333 | 0.150 | | | | |
| | Gender that suffers from any kind of nutrition deficiency | 0.278 | 0.297 | | | | |
| | Gender using fuelwood/firewood as source of energy for cooking | 0.657 | 0.759 | | | | |
| | Gender personally collecting fuelwood for cooking | 0.842 | 0.890 | | | | |
| | Average time gender spent to collect firewood/fuelwood from the forest or bush | 0.336 | 0.356 | | | | |
| | Gender perception about firewood availability 10 years back (less than before) | 0.874 | 0.908 | | | | |
| | Gender used traditional method to cook | 0.612 | 0.729 | | | | |
| Water | Gender that does not have a constant water supply | 0.603 | 0.614 | 0.531 | 0.559 | 6.507 | 0.000 |
| | Gender reporting water conflicts | 0.374 | 0.253 | | | | |
| | Gender that does not have a clean or safe water | 0.545 | 0.604 | | | | |
| | Average time to water source | 0.245 | 0.223 | | | | |
| | Gender that utilizes a natural water | 0.403 | 0.652 | | | | |
| | Inverse of the average number of liters of water stored by each gender | 1.006 | 0.559 | | | | |

**Table 2.** *Cont.*

| Major Components | Sub-Components | Index of Sub-Components | | Major Component Indices | | Two Sample *t*-Test | |
|---|---|---|---|---|---|---|---|
| | | Men | Women | Men | Women | *t*-Value | *p*-Value |
| Natural hazard and climate variability | Occurrence of climate change or variability in the study area | 0.509 | 0.578 | 0.344 | 0.482 | 4.251 | 0.000 |
| | Recorded climate change and climate variability warning received by gender | 0.012 | 0.012 | | | | |
| | Gender with injuries as a result of natural hazards | 0.053 | 0.026 | | | | |
| | Gender that died during climate or natural hazards | 0.024 | 0.012 | | | | |
| | Gender that reported erratic rainfall pattern | 0.576 | 0.725 | | | | |
| | Gender reported reduction in NTFP due to climate change and climate variability | 0.402 | 0.889 | | | | |
| | Gender that reported high temperature and unusual dryness | 0.567 | 0.824 | | | | |
| | Gender that reported erosion destroying farmland | 0.606 | 0.793 | | | | |

Asterisks indicate where there was significant difference between the gender categories (*z*—proportion test) at 95% (*) level of significance.

Adaptive capacity was quantified by the demographic profile of gender, (e.g., percent of female-headed households), the types of livelihood strategies employed, (e.g., predominately agricultural, or collecting natural resources to sell in the market), and the strength of social networks, (e.g., percent of gender assisting neighbors with chores or financial assistance). Last, sensitivity is measured by assessing the current state of a gender's food, water security, and health status. The same sub-components are outlined in Table 1 as well as Equations (2)–(4) were used to calculate the LVI–IPCC. The LVI–IPCC deviates from the LVI when the major components are combined. Rather than merge the major components into the LVI in one step, they are first combined according to the categorization scheme in Table 2 using the following equation:

$$\mathrm{CF_G} = \frac{\sum_{i=1}^{n} WMiMGi}{\sum_{i=1}^{n} WMi} \tag{6}$$

where $\mathrm{CF_G}$ is an IPCC-defined contributing factor (exposure, sensitivity, and adaptive capacity) for the gender $g$, $MGi$ is the major components for gender $g$ indexed by $i$, $WMi$ is the weight of each major component and $n$ is the number of major components in each contributing factor. Once exposure, sensitivity, and adaptive capacity were calculated, the three contributing factors were combined using the following equation:

$$LVI - IPCC_g = (e_{g} - A_g) * S_g \tag{7}$$

where LVI is the Livelihood Vulnerability Index for gender $g$ expressed using the IPCC vulnerability framework, $e$ is the calculated exposure score for gender $g$ (equivalent to the natural hazards and climate variability major component), $A$ is the calculated adaptive capacity score for gender $g$ (weighted average of the socio-demographic, livelihood strategies, and social networks major components), and $S$ is the calculated sensitivity score for gender $g$ (weighted average of the heath, food, and water major components). The LVI–IPCC was scaled from −1 (least vulnerable) to 1 (most vulnerable) [39].

Livelihood Vulnerability Indices Test of Mean Differences

As shown that computation of vulnerability indices would be computed in averages, there is a need for a statistical difference test for the means of the LVI for both gender categories, (i.e., men and women). According to the existing literature, both Mann–White U and the Student's *t*-test have been classified as the most statistical methods in testing for differences in the means of two samples. Ruxton, [40,41] recommended the Mann–White *t*-test to be used for a smaller sample (*N* < 30) with non-formal *t* distribution and unequal population variance. On the other hand, according to Sokal and Rohlf, [42], the Student's *t*-test were best suitable for a larger samples (*N* ≥ 30) where there is an assurance of a homogenous population (equal variance) and normal *t* distribution.

This study adopted the independent two-sample student's test (two-tailed) to test for significant differences in the means of the LVI major components, overall LVI, IPCC vulnerability contributory factors, and the LVI-IPCC indices. The *t*-statistics are calculated using equation

$$t = \frac{(\mu_W - \mu_M)}{\sqrt{\frac{\sigma_W^2}{N_W} + \frac{\sigma_M^2}{N_M}}} \tag{8}$$

where, $\mu F$ and $\mu M$ represent the means computed vulnerability indices for the men and women categories, respectively, $\sigma_W^2$ and $\sigma_M^2$ and $\sigma_M^2$ stands for the standard deviations of the vulnerability indices for the men and women, lastly, $N_W$ and $N_M$ stands for the sample size for the gender categories.

The null hypothesis ($H_0$) for the overall LVI is stated as:

$H_0$ There is no significant difference in the means of the livelihood vulnerability index for men and women categories ($\mu W$ and $\mu M$).

The alternate hypothesis ($H_1$) for the overall LVI is stated as:

$H_1$ There is a significant difference in the means of the livelihood vulnerability index for men and women categories ($\mu W \neq \mu M$).

The same hypotheses were tested for all the LVI major components, the IPCC contributory factors and the LVI-IPCC.

## 3. Results

### 3.1. Men and Women Livelihood Vulnerability Assessment

Ideally, the livelihood vulnerability index (LVI) value ranges between 0 (least vulnerable) and 1 (most vulnerable), and the computed indices for the major components in this study range from 0.121 (least vulnerable) to 0.646 (most vulnerable). The computed vulnerability indices for the major and sub-components and the results of the two-sample *t*-test are presented in Table 2. The result of the two-sample *t*-test indicates a significant difference between men and women categories in all seven examined major components (socio-demographic profile, social networks, livelihood strategies, health, food and nutrition, water, and climate variability). This is presented in Table 2.

The socio-demographic profile, the first major component of vulnerability indices, computed for LVI showed that the women category (0.346) was more vulnerable than the men category (0.320). Meanwhile, the men category was more vulnerable with respect to the age of household-head that also have attended formal school (0.714 and 0.871) than women. A relatively higher percentage of the women (97.87%) have more orphans to cater for than the men (2.13%). The dependency ratios were 0.56 and 0.67 for the men and women categories, respectively (Table 3)

The second major component of the LVI is the livelihood strategies. The vulnerability indices computed indicates that women category was more vulnerable in terms of livelihood strategies (0.646) than men (0.540). The men category has a relatively higher percentage of family members (57.55%) working in the city/foreign country than the women category (42.45%). Slightly above 50% of men category had higher percentage in both growing crops and animal husbandry (52.15%:50.89%) compared to below average (48.75%:49.11%) for women. About (54.05%) of women category collect forest resources and

(65.46%) depend solely on forest resources compared to (45.95%) and (34.54%) for men in forest resources collection and dependency. Though, a small difference exists between men and women regarding agricultural and forest resource dependency, in which both men and women had a very low and equal average agricultural and NTFP livelihood diversification index (0.25). This makes men and women equally vulnerable in the agricultural and NTFP livelihood diversification index.

**Table 3.** Livelihood vulnerability index (LVI) sub-component values with minimum and maximum sub-component values and proportion test for gender in REDD + piloted site/clusters in Cross River State, Nigeria.

| Major Components | Sub-Component | Units | Men | Women | Maximum Values in Selected Villages | Minimum Values in Selected Villages |
|---|---|---|---|---|---|---|
| | | | **Major Component Values** | | | |
| Socio-demographic profile | Dependency ratio | Ratio | 0.56 | 0.67 | 12 | 0 |
| | % of female-headed household | Percent | 27.50 | 72.53 * | 100 | 0 |
| | Average age of female head-household | 1/Year | 0.11 | 0.02 | 0.05 | 0 |
| | % of households where head has attended formal school | Percent | 53.16 | 46.84 | 100 | 0 |
| | % of household with orphans | Percent | 2.13 | 97.87 * | 100 | 0 |
| Livelihood strategies | % of gender with family member working in the city or foreign country | Percent | 57.55 | 42.45 * | 100 | 0 |
| | % of gender growing crops | Percent | 52.15 | 48.75 | 100 | 0 |
| | % of gender raising animals | Percent | 50.89 | 49.11 | 100 | 0 |
| | % of gender dependent solely on forest resources as a source of income | Percent | 34.54 | 65.46 * | 100 | 0 |
| | % of gender collect forest resources from bush, forest or water | Percent | 45.95 | 54.05 | 100 | 0 |
| | Average agricultural and NTFP livelihood diversification index | 1/#livelihoods | 0.25 | 0.25 | 1 | 0 |
| Social networks | Average receive: give ratio | Ratio | 0.71 | 1.50 | 8 | 0.3 |
| | Average borrow: lend money | Ratio | 0.23 | 1.73 | 2 | 0.2 |
| | % of gender that have not gone to the government for assistance | Percent | 31.25 | 68.75 * | 100 | 0 |
| | % of NGO affiliated or belong to | Percent | 69.07 | 30.93 * | 100 | 0 |
| Health | Average time to health facilities | Minutes | 35.88 | 31.67 | 100 | 0 |
| | % of gender with family member down with chronic illness | Percent | 20.44 | 79.56 * | 100 | 0 |
| | % of gender where a family member had to miss work or school in the past 3 weeks due to illness | Percent | 24.59 | 75.41 * | 100 | 0 |
| | % gender family member died due to climate-related hazards | Percent | 42.15 | 57.85 * | 100 | 0 |
| | Average malaria exposure * prevention index | Month*B indicator | 8.17 | 7.8 | 12 | 0 |
| | % of gender that has mosquito net | Percent | 55.31 | 44.69 | 100 | 0 |
| Food and nutrition | % of gender that solely depend solely on NTFP | Percent | 34.54 | 65.46 | 100 | 0 |

**Table 3.** *Cont.*

| Major Components | Sub-Component | Units | Major Component Values | | Maximum Values in Selected Villages | Minimum Values in Selected Villages |
|---|---|---|---|---|---|---|
| | | | Men | Women | | |
| | Average number of months gender having trouble in getting enough food | Months | 7.77 | 7.17 | 100 | 0 |
| | Average crop diversity index | 1/crop | 0.053 | 0.05 | 1 | 0.01 |
| | Average NTFP diversity index | 1/crop | 0.056 | 0.06 | 1 | 0.01 |
| | % of gender that do not save some crops | Percent | 46.15 | 53.85 | 100 | 0 |
| | % of gender that do not save some NTFP resources | Percent | 47.30 | 52.70 | 100 | 0 |
| | % of gender that get sufficient food for the whole year | Percent | 68.96 | 31.04 * | 100 | 0 |
| | % of gender that suffers from any kind of nutrition deficiency | Percent | 44.44 | 55.55 | 100 | 0 |
| | % of gender using fuelwood/firewood as source of energy for cooking | Percent | 46.38 | 53.62 | 100 | 0 |
| | % of gender personally collecting fuelwood for cooking | Percent | 48.60 | 51.40 | 100 | 0 |
| | Average time gender spent to collect firewood/fuelwood from the forest or bush | Minutes | 60.44 | 64.0 | 180 | 0 |
| | % of gender perception about firewood availability 10 years back (less than before) | Percent | 49.03 | 50.97 | 100 | 0 |
| | % of gender used traditional method to cook | Percent | 45.64 | 54.36 | 100 | 0 |
| Water | % gender that do not have a constant water supply | Percent | 49.59 | 50.41 | 100 | 0 |
| | % of gender reporting water conflicts | Percent | 59.68 | 40.32 * | | |
| | % of gender that does not have a clean or safe water | Percent | 47.41 | 52.59 | | |
| | Average time to water source | Minutes | 45.65 | 40.13 | 180 | |
| | % of gender that utilize a natural water | Percent | 38.17 | 61.83 * | | |
| | Inverse of the average number of liters of water stored by each gender | 1/L | 1.0064 | 1.0074 | 1 | 0.0007 |
| Natural hazards and climate variability | % of climate change or variability occurrence in the study the area | Percent | 46.85 | 53.15 | 100 | 0 |
| | % of recorded climate change and climate variability warning received by gender | Percent | 50.00 | 50.00 | 100 | 0 |
| | % of gender with and injuries or as a result of natural hazards | Percent | 66.67 | 33.33 * | 100 | 0 |
| | % of gender that died during climate or natural hazards | Percent | 66.20 | 33.80 * | 100 | 0 |
| | % of gender that reported erratic rainfall pattern | Percent | 44.27 | 55.73 | 100 | 0 |
| | % of gender reported reduction in NTFP due to climate change and climate variability | Percent | 31.14 | 68.86 * | 100 | 0 |

**Table 3.** *Cont.*

| Major Components | Sub-Component | Units | Men | Women | Maximum Values in Selected Villages | Minimum Values in Selected Villages |
|---|---|---|---|---|---|---|
| | % of gender that reported high temperature and unusual dryness | Percent | 40.76 | 59.24 | 100 | 0 |
| | % of gender that reported erosion destroying farmland | Percent | 43.32 | 56.68 * | 100 | 0 |

Asterisks indicate where there was significant difference between the gender categories (*z*—proportion test) at 95% (*) level of significance.

The social networks of the major components of the LVI consist of four sub-components. While 68.75% of women reported not being to any government for assistance in the last 12 months, men (31.25%) reportedly never sought assistance from local, state, or federal government. The computed indices revealed that more women gave assistance and at the same time received from relatives (0.156) compared to (0.053) for men. Additionally, more women (0.850) reported having borrowed more money from friends, families, and relatives than they lent compared to the men category (0.017). Men were more affiliated with NGO or cooperative societies (69.07%) compared to (30.93%) women. The vulnerability index of the social networks major component showed that the women category (0.364) was more vulnerable than the men category (0.121), and there exists a significant difference as indicated by the two-sample *t*-test.

The women category (0.379) seemed to be slightly more vulnerable than their men counterpart (0.331) in terms of health major components of the LVI, as shown by the computed vulnerability indices and the two-sample *t*-test. There are six sub-components that made up the major health component. On average, the men category reported traveling about (35.88) minutes to health facilities, compared to women who traveled 31.67 min.

The women category reported a higher percentage of household members who did not go to school or work for the past 3 weeks due to illness (75.41%), than the men category (24.59%). Furthermore, analysis on gender with the family that has chronic illness was recorded to be high in percentage for women compared to men (79.56% and 20.44%), respectively. A total of 75.41% of women in affirmation said that their family members had missed work or school as a result of illness in the past 3 weeks compared to 24.59% in the men category. Men (0.681) were more vulnerable in terms of the malaria* prevention index than, women (0.650). The percentage of mosquito nets owned by men and women were reported to be 55.31% and 44.69%, respectively. The percentage of recorded death due to climate-related hazards was higher for women (57.85%) than men (42.15%)

The women category (0.507) was more vulnerable to food and nutrition than men (0.465). Above 50% (53.85% and 52.70%) of the women category did not save harvested crops and (non-timber forest product) NTFP resources compared to 46.15% and 47.30% of men that reported not having saved crops and NTFP resources.

The average crop and NTFP diversity index for women was 0.05 and 0.06 compared to 0.053 and 0.056 for men, respectively. Yet, a relatively higher percentage of women depend solely on NTFP resources for survival (65.46%) than men (34.54%). Therefore, the men category was less vulnerable than women in terms of crops and NTFP diversity, especially in a year where there was unsuitable climatic conditions for growing certain crops and availability of expected NTFPs was low. A total of 68.96% of men reported getting sufficient food for the whole year compared to women (31.04%). On average, the number of months of food inadequacy among men and women was 7.77 and 7.17 months per year, respectively.

Women had a higher percentage of nutritional deficiency (55.55%) compared to men (4.44%). Furthermore, (53.62%) of women reported firewood/fuelwood as a source of cooking energy compared to (46.38%) of men. Buttressing this, both men and women reported over 80% collection of firewood/fuelwood by themselves primarily for cooking. The average time of firewood collection for men and women is 60.44 and 64.00 min, respectively. Men (49.03%) and women (50.97%) reported that firewood availability has greatly reduced compared to 10 years back, while 54.36% of women reported using traditional methods to cook compared to 45.64% for the men category. The computed vulnerability indices and the two-sample *t*-test showed that women category was more vulnerable to food and nutrition than men (Table 3).

The sixth major component of the LVI is water, and it consists of six sub-components. Regarding the source of water, (61.83%) of women and (38.17%) of men agreed that they utilized natural water such as rivers, rains, lakes, dams, and streams. About 59.68% and 40.32% of men and women, respectively, reported water conflicts. Similarly, the average time for men and women to travel to the water sources was (45.65) and (40.12) minutes, respectively. Women stored about 135.33 L on average compared to 157 L for men. Furthermore, (50.41%) of women had a constant supply of water compared to (49.492%) of men. Finally, about (52.59%) of women also reported that they do not have access to clean or safe water compared to men (47.41%). When all the six sub-components indices were averaged, the women category was significantly more vulnerable to the water components (0.559) than men (0.531).

The results of the two-sample *t*-test revealed that there is a significant difference in the indices of climate variability, and the unexpected climate change major components of the LVI. According to the computed indices, the women category reported higher indices in climate change occurrence (0.578), erratic rainfall (0.725), reduction in NTFPs (non-timber forest products) due to climatic variability (0.889), high temperature, (0.824), and reported erosion destroying farmland (0.793). Higher death and injuries percentages were recorded in men (66.20%:66.67%) than women (33.80%:33.33%), respectively. However, there was a consensus between men and women in all communities studied that they received climate change and variability warning at 50%. (Table 2). Consequently, women were more vulnerable on the score (0.482) than men (0.344), respectively, as also revealed by the result of the two-sample *t*-test (Table 4).

**Table 4.** LVI-IPCC computed indices with contributing factors and two-sample *t*-test results for gender categories in REDD + piloted site/clusters in Cross River State, Nigeria.

| Contributory Factor | Computed Index | | Two—Sample *t*-Test | |
|---|---|---|---|---|
| | **Men** | **Women** | ***t*-Value** | ***p*-Value** |
| Exposure | 0.344 | 0.482 | −10.576 | 0.000 |
| Sensitivity | 0.463 | 0.489 | 9.753 | 0.000 |
| Adaptive capacity | 0.364 | 0.462 | 8.974 | 0.000 |
| LVI-IPCC | −0.0093 | 0.0098 | 2.581 | 0.002 |

$LVI - IPCCe_{Men} = (e_{Men-}A_{Men}) * S_{Men}$ = (0.344 − 0.364) * (0.463) = −0.0093. $LVI - IPCCe_{women} = (e_{Women-}A_{Women}) * S_{Women}$ = (0.482 − 0.462) * (0.489) = 0.0098. LVI: men −0.0098. LVI: women 0.0098.

When all the seven major components of the LVI were aggregated, the women category with an overall LVI of 0.618 was considered to be more vulnerable to climate change and variability than the men category with an overall LVI of 0.509.

The results of the independent two-sample *t*-test revealed a significant difference between the computed LVIs of the women and men categories (Table 3).

*3.2. Gender and Proportionality Assessment*

Just 27.5% of the men agreed that their households were headed by women. This small percentage indicates that the men do not see women being the household head a significant influence in decision-making particularly in African settings. Proportion test results found this difference to be highly significant ($z = -15.245$, $p = 0.000$). Thus, it can be stated with nearly above 50% confidence that both men and women categories have a vastly different views concerning who should be the household head and this has a great effect on socio-demographic contribution to climate change and variability impacts. About 98% of the women category hold the opinion that being a household head and raising orphans contributed immensely to socio-demographic profile, which will put women in a tight corner in climate change resilience compared to 30% of the men category. However, attending formal school for each gender was not statistically significant (Table 2).

Livelihood strategies, a major component according to this study, shows the difference between gender and different view of livelihood strategies. Men had a higher percentage (57.55%) compared to (42.45%) women (family members working in the city), and the majority of women (65.46%) to men (34.54%) were significant in difference at *z*-test proportion results test ($z = 14.201$, $p = 0.000$; $z = 10.451$, $p = 0.000$). The highest proportion from sub-components contributing to livelihood and sustenance shows that both groups believe these two variables cannot be over-emphasized as far as climate change and livelihood vulnerability is a concern. On the other hand, about 70% of the women feel that the "*Not going to government*" positioned them to be vulnerable to climate change impacts in terms of social networks. Though, fewer than one-third of men also agreed on the sub-component that no government support had a great contribution to vulnerability from social networks perspectives, (The *z* value for the proportion test is 10.240, with a *p* value of 0.0000. The women perceive that differences exist to a much greater degree with higher confidence.)

Just above 60% of the men category have the impression that belonging to society or NGO bodies contribute to social networks in order to be less vulnerable to climate change impact. Even among the women category, the majority (79.56%) of family members were down with chronic illness compared to men (20.44%) as a sub-component in contributing to the major component in determining gender vulnerability. This proportion produced a *z* value of 8.243 and it is statistically different at the 0.000 significance level. It can be stated with nearly 100% confidence that women are more likely than men to be affected by the shock of climate change and variability. The percentage of women whose family has missed school to illness and died due to climate hazards was 74.41%:57.85% compared to men at 24.44%:42.15%, respectively. The result suggests that health-wise, in relation to climate change vulnerability, women perceived these sub-components as an important factor in climate change impacts and vulnerability negotiation.The percentage proportion among the women is higher than 50% across all the sub-components groups for food and nutrition except "getting *sufficient food for the whole year*" with men (68.96%) with *p*-value of 0.001, and it is significantly different between the gender group. Both gender categories hold a very similar view in terms of obtaining a constant water supply and unclean water, with the exception to water conflicts and natural water utilization which is significant with a *z* score of 7.907, $p = 0.000$, providing higher confidence that women and water-related issues can never be over-emphasized as far as climate change vulnerability is concerned. Men and women categories were in perfect agreement regarding the climate warning received and the occurrence of climate variability, and shock in the study area as related to natural hazards and climate variability. Above 60% and 35% (men and women, respectively), believe injuries and death as a sub-component contributed to their vulnerability in terms of climate variability. Statistical testing revealed the difference between men's and women's opinions ($z = 18.421$, $p = 0.000$). A significant difference was found between the perception of men and women concerning erratic rainfall (men 44.27%:women 55.73%), non-timber forest product reduction due to climate variability (men 31.14%:women 68.68%), high temperature (men 40.76%:women 59.24%), and farmland destruction due to erosion (men 43.32%:women 56.68%). The *z* score of 7.327 indicates a significant difference between the

views on gender and natural hazards and climate variability. Overall, the shared responses from both women and men indicate that all the sub-components are fundamental factors that contributed to climate change-induced vulnerability (Table 2).

The computed vulnerability indices of the major components of the LVI and the overall LVI for men and women categories are presented in the gender vulnerability radar diagram in Figure 3.

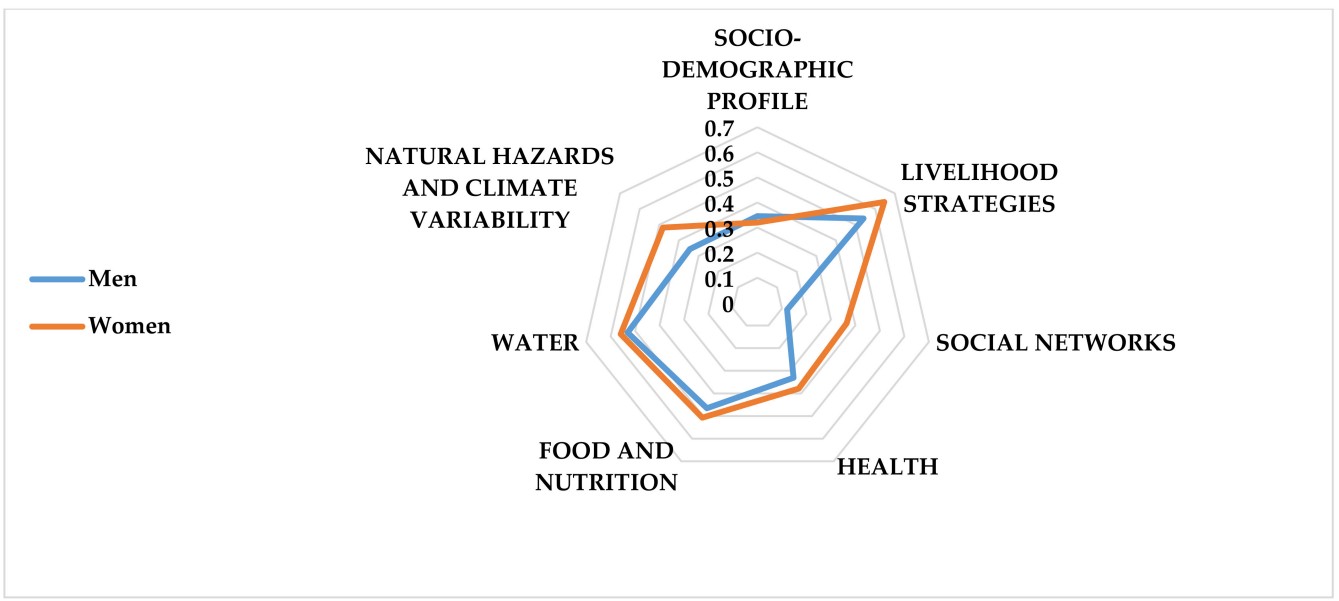

**Figure 3.** Vulnerability spider diagram of the major components of the livelihood vulnerability index (LVI) for men and women in REDD + site/clusters, Cross River, State, Nigeria.

Figure 3 showed that the women category is more vulnerable in terms of livelihood strategies, social networks, health, food and nutrition, water and related natural hazards, and climate change and variability profile.

Based on computed vulnerability contributory factor indices (CFI), the women category was more vulnerable than men in all the three factors in terms of EXPOSURE (women, 0.482, men 0.344), SENSITIVITY (women 0.489, men 0.463) and ADAPTIVE CAPACITY (women 0.462, men 0.364) variability and natural hazards than men (0.344).

The computed LVI-IPCC specifies that the women category was more vulnerable to climate change and variability (0.0098) than the men category ($-0.0098$). Furthermore, the result of the independent two-sample student *t*-test showed that there were significant differences in the means of the LVI-IPCC vulnerability contributory factors for women and men categories in REDD + piloted site/clusters (Table 4). Consequently, ($H_0$) was rejected. The implication is that even though the two categories of the group examined were residing in the same geographical location, the women category was more exposed and sensitive to climate change and variability with the least adaptive capacity. These contributory factors have positioned them to be more vulnerable.

## 4. Discussion

### 4.1. Men versus Women LVI Comparison and Its Practical Implications

The information that contributes most by gender category to climate change vulnerability in REDD + piloted site/clusters in Nigeria, was exclusively provided by the major components as presented in Figure 3. The information is similar to several other studies on the IPCC-LVI vulnerability assessments [43–46]. This study has shown the consistent vulnerability of women to climate change impact compared to men in agriculture, natural resources management, and natural hazards/disasters prone areas across the globe. This information can be created for individuals, households, and communities for further sup-

port. Women are more vulnerable with a relatively larger percentage of orphans to cater for. The dependency ratio of women is also higher than men. This indicates that many people or other groups were dependent on the labor of others to survive or makes the ends meet [47]. Furthermore, the higher proportion of men that have formal education compared to women shows that in terms of education, women are still lagging behind. (The actual percentage of female-headed households in this study was 27.50 compared to the study by Milazzo and van de Walle [48] for Nigeria. Reference to water as a major component in this study, Nigeria suffers from recurrent epileptic rainfall and high dryness [49], different plastic water storage containers (about 500–1000 L capacity), which were installed with boreholes in the selected communities where the data were collected was observed. It will be a great advantage if these management practices were efficiently worked upon more, consequently, this might be the turning point for women/females to have access to community pumped water, rather than using natural water sources that might sometimes be contaminated or have the capacity to cause waterborne diseases such as cholera. This also suggests that the time and energy expended in looking for natural water sources could be channeled into a more meaningful and productive role and responsibility both in the household and the community at large. This corroborates the study of Nounkeu et al. [50], that women played a key role in water fetching and its daily management, which is limited, and access can affect their ability to care for themselves, their children, and productive purposes. However, the non-availability of clean and safe water recorded by the women category can be attributed to chore and productive responsibilities that require more water "sensitivity" roles such as drinking, bathing and cooking compared to men. This resonates with Radonic and Jacob [51], that water collected by women is actually not clean and pure with associated physical burden, which encompassed the effects on the body during collection and management as well as the effects of contact with contaminated water such as skin irritations, discolorations, hair dryness and hair loss. Men experienced more water-related conflicts in the study area compared to the women group. It was suggested that the basis of water-related conflicts in the study area were attributed to water sharing and degree of usage by men and women, respectively. Water fetched for the household by women is primarily for domestic activities and selective animal husbandry, while men fetched water for big water consumption activities such as irrigation [52].

Similarly, a higher proportion of women depended solely on (non-timber forest produce) NTFPs, crop, and NTFPs diversity indexes, among others, which is an indication that their livelihood strategies were more of climate-dependent occupations such as agriculture [53]. The cultivation of a small portion of infertile land and climatic variability will produce low food output that cannot be depended on for the whole year [54]. Consequently, women use ecosystem services such as collecting wild nuts, mushrooms, and spices, mainly for home consumption and surplus for sale, which serves as an alternative strategy for livelihood enhancement [54]. Additionally, both men and women categories did not report the same level of food insecurity in terms of (1) food sufficiency, (2) food struggling, (3) nutrition deficiency, (4) energy collection, usage, and fuelwood perception over 10 years. Men were more engaged in seed storage and other food management practices than women, which is the reason for the higher percentage in not saving both crops and NTFP resources. This suggests that awareness and education related to crop and NTFP storage, management, and preservation techniques might constitute an appropriate innovation for women considering their current state of food insecurity status compared to the men category [55]. The higher vulnerability suggests that timely and effective gender education [55] that will target food saving, dependency on crops, and NTFPs with sustainable use of forest resources might be an appropriate intervention for women in (Reducing Emission from Deforestation and Forest Degradation+) REDD + piloted site/clusters despite high current of food and nutrition status.

Higher diversification of sources of income beyond farming was reported by the men category. This includes the raising of livestock such as fowls, pigs, goats, and, in a few cases, dogs. Despite these practices, women were more vulnerable than men in terms of

the livelihood strategies index [56]. Another livelihood coping strategy adopted by the men category in the study site was to travel or migrate in looking for greener pastures [57]. The migration pattern has been reported to meet the immediate income needs of the family but has indirectly exacerbated the existing responsibilities hanging on the neck of the women [58]. Therefore, this is the reason why the higher score was recorded for the family members who reported working or migrating outside the community for greener pastures. Additionally, women's strong affinity and closeness to forest resources make them have higher NTFP collection and agricultural livelihood index [59]. This will serve as a buffer and livelihood enhancement in the face of climate change impacts.

Although men reported a longer average time to health facilities and higher malaria exposure with bednet ownership, the women category was more with family members with a higher prevalence of chronic illness and death related to climate hazards. The higher percentage of respondents from the women category who were so sick and have either missed school or work in the past 3 weeks compared to the men could be attributed to the larger percentages of able men who reside outside the community, leaving behind predominantly women, minors, and orphans to be taken care of [60]. With these findings, it is suggested that malaria may pose a negative threat to gender income and their respective working days. This result resonates with Gunda et al. [61], in which households spent an average of USD 56.60 on complicated malaria cases, with an average loss of eight productive working days and an average 24% of the household monthly income lost in rural households of the Gwanda district of Zimbabwe.

Based on these findings, women must be more targeted with bednet distribution and close monitoring of any related health assessment due to a greater number of children, minors, and the under-aged who are taken care of [62]. Consequently, determination of diseases causing people to miss school or work can be easily identified and to propose a way out for the policy maker in the study sites.

The main reason for creating borrow money: lend money and receive assistance: give assistance ratio was to measure the degree or extent to which gender rely on family members, friend, associates, and loved ones for in-kind help and financial or monetary help in time of hardship or needs [63]. An assumption was made that the gender that receives money or in-kind assistance from time to time but offers little assistance to others in the community is more insecure and vulnerable compared with those that are affluent and have time to help others.

The findings that women had a higher borrow: lend and receive: give ratio may be attributed to their availability in the community. Furthermore, it might also be due to the village structure of the explored communities which is extended in nature. These living arrangements may have influenced the examined help versus obligation, but this is beyond the scope of this study. The result corroborates the study of Phan et al. [64], that regarding the gender social networks, the higher the involvement of female-headed households in community activities, and their associations with local organizations, the more support they receive, because their affiliation with these organizations provides them with more opportunities to access to information, communication, services, and interventions to improve their livelihoods and well-being.

Creating community bonds and high levels of trust among communities to helps to adapt to the impacts of climate change.

However, there is tendency for food security and health indicators to be more flexible or easy to measure than social characteristics. Although the combination of social indicator sub-components indexes did not contribute much to the LVI for both the men and women gender categories, this could be due to selected or chosen indicators, not a true reflection of local and social customs of the community. Despite the challenges faced in qualifying social networks, their inclusion in climate change vulnerability assessment is very important because the majority of the adaptation plans and behaviors rely on collective insurance mechanisms such as thrifts and cooperative society.

Finally, although women recorded a higher absolute number of natural hazards (five out eight) sub-components over past years, the number of climate warnings received was equal for both men and women. This corroborate Ngigi et al. [65] that there are gender-specific preferences of information dissemination channels such as accessing agricultural and climate information through group-based approaches, neighbors, meetings with local leaders and extension officers etc. for climate change adaptation among rural households in Kenya. Among indexes that contributed to higher natural hazards and climate variability (NHVC), scores for women include high temperature, erratic rainfall, destruction of farmland with erosion, and early warning system. Consequently, a planned early warning system for men and women as individuals and the community at large may help genders prepare for unexpected extreme weather events. An association of local farmers and forest resource gatherers equipped with seasonal and effective weather forecasts would help the farmers with timely planting, NTFP abundance season, and management of scarce water available for irrigation purposes. It is most prudent to have prior information on weather forecasts that will enable farmers to adequately adapt to changing conditions [65]. Consequently, the findings of this study resonate with that of Naab et al. [66], and Basiru et al. [67] who revealed that female-headed households were more vulnerable to climate change due to low adaptive capacity in eastern part of Ugandan and forest-based community in Southwest Nigeria respectively.

With reference to the gender and vulnerability dimension in this study, it is clearly shown that power relations, water and health accessibility, and political, social, and economic structure were all reflected. Therefore, this implies that vulnerable groups would need some temporary assistance to recover when hit by climate change and variability, natural hazards such as floods, and any other form of shock.

It is anticipated that the LVI would be helpful to policymakers when evaluating livelihood vulnerability to climate change impacts within the REDD + piloted communities in which they drew their livelihood and to develop programs to strengthen the currently vulnerable sector in the area.

### 4.2. Benefits and Research Implications

By changing the value of the indicator that predicted the change and recalculating the total vulnerability index, the LVI and LVI–IPCC could be used to assess the impact of a program or policy. If the purpose of a water sector intervention is to reduce the time taken to reach a community's principal water supply, for example, the desired travel time may be factored in and a new LVI determined. The new LVI might then be compared to the baseline LVI to determine the intervention's impact on climate vulnerability in the community. Similarly, under simple climate change scenarios, the LVI might be used to estimate future vulnerability. To measure the specified sub-components, the LVI and LVI–IPCC use gender disaggregated primary data in the household. As a result, this strategy avoids the drawbacks of secondary data-driven methodologies, such as the effects of mixing data acquired at different temporal and/or spatial scales and for different reasons. Furthermore, with the LVI approach, sources of measurement error are limited to our household survey methods and self-reported data error. Researchers who rely on secondary data, on the other hand, frequently lack information on measurement error and hence have no method of estimating potential biases in findings interpretation. Furthermore, this study was able to show that in resource-scarce settings, high-quality home survey data with low missing answer frequencies may be collected. As a result, the LVI method aids in the circumvention of the missing data problem that plagues many secondary data sources. Finally, the LVI's sub-components and weighting structure can be customized to meet the demands of a specific community or end-user, and as other assessments have been discussed above, these features can be fixed within policy maker assessment frameworks. The implication of this result on the REDD + design and implementation in Nigeria is that it will assist REDD + (apart $CO_2$ mitigation) as a potential means to better protect forests and strengthen the livelihoods of the people that depend on them in terms of identification of deforestation drivers; par-

ticipation, consultation, and stakeholder engagement; forest monitoring systems; forest conservation, governance, social and environmental risks/safeguards according to Nigeria REDD + statement and recommendation. Therefore, REDD + implementation focused on:

(i)　　Proper engagement of forest community, training, and full inclusiveness throughout the program's implementation;

(ii)　　Broad approach that goes beyond forest conservation to address questions of land management, afforestation and reforestation, ecosystem restoration, sustainable agriculture, and community-based gender livelihoods;

(iii)　Workable gender resilience and adaptive capacity;

(iv)　Benefit-sharing mechanisms and community conflict, providing guidance on how to address them in the context of gender and REDD + project.

*4.3. Limitation of the Study*

Unconventional approaches for estimating the comparative vulnerability of communities to climate change impacts include the LVI and LVI–IPCC. Each method provides a complete picture of the factors that contribute to gender livelihood vulnerability in a specific area. To reach a wide range of users, the methods for computing the LVI and LVI–IPCC were meant to be simple. When two or more research areas are compared using vulnerability spider and triangle diagrams, more information can be collected. Bias in selecting sub-components, the relationship between the sub-components, and an assumption of equal weights for all components were all the limitations of this study.

**5. Conclusions**

This study clearly showed that the gender livelihood vulnerability dimension can be analyzed using social, human, physical, financial, and natural capital within the context of the sustainable livelihood framework. This will aid the easy identification of the most vulnerable groups in climate change adaptation programs such as REDD +.

The findings of this study demonstrated a discrepancy in men's and women's sensitivity to climate change and variability in REDD + piloted sites, in Cross River State, Nigeria. Furthermore, findings also showed that women are more vulnerable to five capitals (social, human, physical, financial, and natural capital) in terms of food, health, social networks, water, socio-demographic profile, natural hazards, and climatic variability major components.

Overall, women were much more exposed to climate change and variability than men. It is concluded that a gender lens may be the best instrument along which differences in vulnerability to climate change and variability impacts can be assessed. The unintended consequence of neglecting gender course in climate change debates can harm or misinform the direction of adaptation planning, due to the wrong assumption in allocating resources about the vulnerability outcomes. The study also concluded that reducing vulnerability among vulnerable women's groups requires that more efforts be put into a proactive adaptation that will address exposure, sensitivity, and adaptive capacity indicators. Furthermore, the findings of the research can help understand the contributing factors to vulnerability and enhance gender adaptive capacity in terms of income and livelihood diversification options at a community level.

Based on these findings, the study recommends that women be given priority in both ongoing and new climate change and agriculture intervention projects by empowering them to venture into other income-generating activities. This will serve as the best way to diversify their sources of livelihood and increasing their resilience to climate change and variability.

Finally, gender vulnerability assessment in climate adaptation programs such as REDD + could be a pathway to achieving Nigeria's National Environmental, Economic and Development Study (NEEDS) for Climate Change and gender equity goal ("*gender equality and social inclusion should be mainstreamed and REDD + activities and benefits should reach communities equitably*"). However, the total exclusion of the men category in climate

change vulnerability in intervention programs such as REDD + should be carefully guided against. The top priority is to give women the chance to participate and make decisions in such programs.

In order to analyze livelihood vulnerability to climate change from a gender perspective, the study demonstrates the flexibility and appropriateness of calculating LVI. The study's findings, however, are limited to the vulnerabilities that existed at the time of the research and are majorly applicable to the study's focus. Future studies might adapt the LVI for appropriate application in actual circumstances and a broadened focus.

Gender and intra-household differences are not examined in this study; therefore, using both quantitative and qualitative techniques in future studies may be considered as the best approach.

**Author Contributions:** Conceptualization, A.O.B., A.O.O., O.O.A. (Olubusayo Omotola Adekoya) and F.C.; methodology, A.O.B., A.O.O., O.O.A. (Olubusayo Omotola Adekoya) and F.C.; software, A.O.B. and A.O.O.; validation, A.O.B., V.O.O. and O.O.A. (Opeyemi Oluwaseun Awodutire); formal analysis, A.O.B., L.A.A., E.K.A. and F.C.; investigation, A.O.B., O.O.A. (Olubusayo Omotola Adekoya) and A.O.O.; resources, F.C. and V.O.O.; data curation, O.O.A. (Opeyemi Oluwaseun Awodutire); writing—original draft preparation, A.O.B., F.C. and L.A.A.; writing—review and editing, A.O.B., V.O.O. and E.K.A.; visualization, O.O.A.; supervision, L.A.A. and F.C.; project administration, A.O.B. and O.O.A. (Opeyemi Oluwaseun Awodutire); funding acquisition, A.O.B. and A.O.O. All authors have read and agreed to the published version of the manuscript.

**Funding:** This research was funded by the UK Research and Innovation (UKRI) through the Global Challenges Research Fund (GCRF) program, Grant Ref: ES/P011306/under the project Social and Environmental Trade-offs in African Agriculture (SENTINEL) led by the International Institute for Environment and Development (IIED) in part implemented by the Regional Universities Forum for Capacity Building in Agriculture (RUFORUM) and African Forest Forum (AFF), World Agroforestry Centre (ICRAF), Nairobi, Kenya.

**Institutional Review Board Statement:** Not applicable.

**Informed Consent Statement:** Not applicable.

**Data Availability Statement:** Not applicable.

**Acknowledgments:** We acknowledged the useful contribution and support from the Cross River Ministry of Forestry. Special thanks to: Clement Adetogun, Asuquo Francis, Agana, Kunle Sodiya, Akinwale Gabriel, Adegoke Adekola Felix, Godwin Kowero, Doris Mutta, Louise Avana, Daud Kachamba, Adesuyi Adekunle, Majaliwa Nwanjalolo and the staffs of Department of Forestry and Wildlife Management, Federal University of Agriculture, Abeokuta, Nigeria for all the support. SENTINEL teams from Africa and UK such as David Ekepu, Adam Devenish, Berth Downe, Geoff Griffiths, and Anthony Egeru—you are greatly appreciated.

**Conflicts of Interest:** The authors declare no conflict of interest.

## Appendix A

**Table A1.** Major components and sub-components comprising the livelihood vulnerability index (LVI) developed for gender in REDD + piloted site/clusters in Cross River State, Nigeria.

| Major Components | Sub-Components | Explanation of Sub-Components | Survey Question | Source |
|---|---|---|---|---|
| Socio-demographic profile | Dependency ratio | Ratio of the population under 15 and over 19 and 64 years of age to the population between 19 and 64 years of age | Could you please list the ages and sexes of every person who eats and sleeps in this house? If you had a visitor who ate and slept here for the last 3 days, please include them as well. | Adapted from Hahn et al. [38] |

**Table A1.** *Cont.*

| Major Components | Sub-Components | Explanation of Sub-Components | Survey Question | Source |
|---|---|---|---|---|
| Livelihood strategies | Average gender age of female Household Head | Average age of household that happens to be female | As a female household head, what is your age? | Purposefully developed for this questionnaire |
| | % of female-headed households | Percentage of household where the primary adult is female. If a male head is away from the home > 6 months per year the female is counted as the head of the household | Are you the head of the household? | Adapted from Hahn et al. [38] |
| | % of gender where the head of household have formal education | Percentage of genders where the head of the household reports that they have attended formal school. | Did you ever go to school? | Adapted from Hahn et al. [38] |
| | % of gender recorded orphans less than 13years of age | Percentage of gender that have at least 1 orphan living in their home. Orphans are children <13 years old who have lost one or both parents. | Are there any children less than 13 years old from other families living in your house because one or both of their parents has died? | Adapted from Hahn et al. [38] |
| | % of gender depend solely on forest resources as source of income | Percentages of gender whose gathering of forest resources determines their source of income | Do you depend sole on forest resources as your source of income? | Purposefully developed for this questionnaire |
| | % of households with family member working in a city or foreign country | Percentage of genders that report at least 1 family member who works outside of the community for their primary work activity | How many people in your family go to the city ior foreign country to work? | Adapted from Hahn et al. [38] |
| | % of gender raising animals | Percentage of gender that report raising animals for livelihood enhancement | Do you raise animals to support your livelihood? | Purposefully developed for this questionnaire |
| | % of gender growing crops | Percentage of gender that report growing crops for livelihood enhancement | Do you grow crops to support your livelihood? | Purposefully developed for this questionnaire |
| | % of gender collect forest resources from bush, forest or water | Percentage of gender that reported collecting forest resources for income and consumption | Do you grow collect natural resource from bush, forest or water? | Purposefully developed for this questionnaire |
| | Livelihood diversification index (range: 0.25–1) [a] | The inverse of (the number of agricultural and NTFPs livelihood activities +1) reported by a genders, e.g., an individual that farms, raises animals, and collects natural resources will have a livelihood diversification index = 1/(3 + 1) = 0.25. | Same as above | Adapted from Hahn et al. [38] |
| Health | Average time to health facility (minutes) | Average time it takes the genders to get to the nearest health facility. | How long does it take you to get to a health facility? | Adapted from Hahn et al. [38] |

**Table A1.** *Cont.*

| Major Components | Sub-Components | Explanation of Sub-Components | Survey Question | Source |
|---|---|---|---|---|
| | % of gender with family member with chronic illness | Percentage of gender that report at least 1 family member with chronic illness chronically ill (they get sick very well, chronic illness was defined subjectively by often) | Is anybody in your family with chronic illness? | Adapted from Hahn et al. [38] |
| | % of gender where a family member had to miss work or school in the last 3 weeks due to illness | Percentage of gender that report at least 1 family member who had to miss school of work due to illness in the last 3 weeks | Has anyone in your family been so sick in the past 3 weeks that they had to miss work or school? | Adapted from Hahn et al. [38] |
| | % of gender recorded family death due to climate-related disaster | Percentage of family member climate-related disaster have claimed their lives | Did any members of your family have died due to any climate-related disaster? | Adapted from Hahn et al. [38] |
| | Average malaria exposure prevention index (range: 0–12) | Months reported exposure to malaria owning at least one bednet indicator | Which months of the year is malaria particularly bad? | Adapted from Hahn et al. [38] |
| | % of gender that has mosquito net/bednet | Percentage of gender with ownership of net/bednet | Do you have mosquito net? | Adapted from Hahn et al. [38] |
| Social networks | Average receive: give ratio (range: 0–15) | Ratio of (the number of types of help received by gender in the past month + 1) to (the number of types of help given by gender to someone else in the past month + 1). | In the past month, did relatives or friends help you and your family: (e.g., obtain medical care or medicines, sell animal products or other goods produced by family, take care of children) In the past month, did you and your family help relatives or family? | Adapted from Hahn et al. [38] |
| | Average borrow: lend money ratio (range: 0.5–2) | Ratio of gender borrowing money in the past month to gender lending money in the past month, e.g., if gender borrowed money but did not lend money, the ratio = 2:1 or 2 and if they lend money but did not borrow any, the ratio = 1:2 or 0.5. | Did you borrow any money from relatives or friends in the past month? Did you lend any money to relatives or friends in the past month? | Adapted from Hahn et al. [38] |
| | % of gender that their family have not gone to their local government for assistance in the past 12 months | Percentage of gender that reported that they have not asked their local government for any assistance in the past 12 months | In the past 12 months, have you or someone in your family gone to your community leader for help? | Adapted from Hahn et al. [38] |
| | % of NGOs gender affiliated with | Percentage of NGOs the gender category belongs to in the community | Do you belong to any affiliated body or NGOs? | Purposefully developed for this questionnaire |
| Food and nutrition | % of gender that obtain sufficient food for the whole year | Percentage of gender that experience availability of food throughout the year | Do you have sufficient food throughout the year? | Purposefully developed for this questionnaire |

**Table A1.** *Cont.*

| Major Components | Sub-Components | Explanation of Sub-Components | Survey Question | Source |
|---|---|---|---|---|
| | Average number of months gender struggle to find food (range: 0–12) | Average number of month's gender struggle to obtain food for themselves. | Do you have adequate food the whole year, or are there times during the year that you do not have enough food? | Adapted from Hahn et al. [38] |
| | % of gender depend solely on non-timber forest products (NTFPs) | Percentage of gender that depend solely on NTFPS gathering for consumption and income | Do you collect NTFPs for consumption and income? | Purposefully developed for this questionnaire |
| | Average crop diversity index (range: >0–1) [a] | The inverse of (the number of crops grown by gender +1), e.g., an individual that grows pumpkin, maize, okra beans, and cassava will have a crop diversity index = 1/(4 + 1) = 0.20. | What kind of crops do you grow? | Adapted from Hahn et al. [38] |
| | Average NTFP diversity index (range: >0–1) [a] | The inverse of (the number of NTFP by gender +1), e.g., an individual that collected leaves, snails, rattan, mushroom, and fruits will have a NTFP diversity index = 1/(4 + 1) = 0.20. | What kind of NTFPs do you collect? | Purposefully developed for this questionnaire |
| | % of gender that do not save crops | Percentage of gender that do not save crops from each harvest. | Do you save some of the crops you harvest to eat during a different time of the year? | Purposefully developed for this questionnaire |
| | % of gender that do not save NTFPs | Percentage of gender that do not save NTFPs resources from year to year. | Do you save some NTFPs resource again next year? | Purposefully developed for this questionnaire |
| | % of gender that suffers from any kind of nutritional deficiency | Percentage of gender that suffers any kind of nutritional deficiency | Do you suffer any nutritional deficiency? | Purposefully developed for this questionnaire |
| | % of gender source of energy for cooking especially firewood | Percentage of gender uses firewood as source of energy for cooking | Do you use firewood as source energy for cooking? | Purposefully developed for this questionnaire |
| | % of gender collecting fuelwood for cooking | Percentage of gender collecting fuelwood for cooking | Do you collect fuelwood for cooking only? | Purposefully developed for this questionnaire |
| | Average time gender spent to collect firewood/fuelwood from the forest | Average time an individual spent in collecting fuelwood/firewood from the forest | How many minutes do you spent on fuelwood collection from forest? | Purposefully developed for this questionnaire |
| | % of gender ascertain firewood availability has reduced in the last 10 years | Percentage of gender perception of fuelwood availability over 10 years | What is the availability of firewood in the last 10 years in this community? | Purposefully developed for this questionnaire |
| | % of gender using traditional methods of cooking | Percentage of gender using traditional methods for cooking | Which methods do you use in cooking your food? | Purposefully developed for this questionnaire |
| Water | % of gender that do not have a constant water supply | Percentage of gender that suffers inconsistent water supply in the community | Do you have constant water supply? | Adapted from Hahn et al. [38] |
| | % of gender that do not have a clean or safe water | Percentage of gender that doesn't have access to clean or safe water in the community | Do you have access to clean or safe water? | Purposefully developed for this questionnaire |

<p style="text-align:center">**Table A1.** *Cont.*</p>

| Major Components | Sub-Components | Explanation of Sub-Components | Survey Question | Source |
|---|---|---|---|---|
| Natural hazards and climate variability | % of gender reporting water conflicts | Percentage of gender that report having heard about conflicts over water in their community | In the past year, have you heard about any conflicts over water in your community? | Purposefully developed for this questionnaire |
| | Average time to water source (minutes) | Average time it takes gender to travel to their primary water source. | How long does it take to get to your water source? | Adapted from Hahn et al. [38] |
| | % of gender that utilize a natural water source | Percentage of gender that report a creek, river, lake, pool, or hole as their primary water source. | Where do you collect your water from? | Adapted from Hahn et al. [38] |
| | Inverse of the average number of liters of water stored per gender | The inverse of (the average number of liters of water stored by each gender + 1). | What containers do you usually store water in? How many? How many liters are they? | Purposefully developed for this questionnaire |
| | % of climate change and variability occurrence in the study community as reported by gender category | Percentage of time climate change have affected the study community | How many times have you experienced climate change extreme event in this area? | Purposefully developed for this questionnaire |
| | % of gender that received a warning about the pending natural disasters | Percentage of gender that received a warning about the most severe flood, drought and high temperature in the past 10 years | Did you receive a warning about the flood/erosion/drought before it happened? | Purposefully developed for this questionnaire |
| | % reporting death of person or family member | Percentage of gender reported any death due to climate-related hazards | Has anyone of your family died of any climate-related hazards? | Purposefully developed for this questionnaire |
| | % reported injuries during extreme event | Percentage of gender reported injuries due to extreme weather events | Have anyone of your family member injured during extreme weather event? | Purposefully developed for this questionnaire |
| | % reported erratic rainfall pattern | Percentage of erratic rainfall pattern recorded over last 10 years | Have you been experiencing erratic rainfall pattern in this area for the last 10 years | Purposefully developed for this questionnaire |
| | % of gender that reported reduction in NTFPs resources due to climate variability | Percentage of gender reported disappearance of forest resources due to climate variability | What is the status of NTFPs resource in accordance with climate variability for the past 10 years in this community? | Purposefully developed for this questionnaire |
| | % reported high temperature | Percentage of gender reported extreme high temperature | Has the level of temperature increases in this community over last 10 years? | Purposefully developed for this questionnaire |
| | % reported destruction of farmland and properties by erosion | Percentage of gender reported erosion have destroyed their farmland and properties | Has erosion destroyed your farmland or properties before? | Purposefully developed for this questionnaire |

[a] Some indicators such as the Livelihood Diversity index were created because of an increase in the crude indicators, i.e., the number of livelihood activities undertaken by gender decreases vulnerability. As a result, inverse of these were taken by creating numbers that reflects lines of reasoning, thereby assigning higher values to gender with a low number of livelihood activities.

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
