# Peer review of "Livelihood Vulnerability Index: Gender Dimension to Climate Change and Variability in REDD + Piloted Sites, Cross River State, Nigeria"

_land, doi:10.3390/land11081240_

Round 1

Reviewer 1 Report

Dear authors,    Thanks for the opportunity to review your manuscript.    This manuscript can be considered for publication after some MINOR revisions. The manuscript requires MINOR REVISION before it is considered for publication. Thus, the authors are strongly recommended to revise the manuscript accordingly and subsequently submit a revised work. The manuscript requires further effort in order to be considered for publication. It requires further MINOR revision and amendments.    My specific comments are presented as follows:   1. The manuscript is well written in an engaging and lively style on one of the very interesting topics nowadays, GENDERED VULNERABILITY PATTERNS OF HOUSEHOLDS. The level is quite appropriate to the LAND journal's readership.   2. The authors are hereby suggested to revise the referencing format. There is so much inconsistency in the references list. Some references have DOI numbers, some don't have.    3. The conclusion section is too much. Almost two pages. That's too much. Please, kindly reduce it to half a page.   4. The paper would benefit from stylistic changes to the way it has been written for a stronger, clear, and more compelling argument. There are a few sentences that need rephrasing and clarity.    All the best of luck!

Reviewer 2 Report

Major and minor comments:

Title:

The title does not say anything about the vulnerability, vulnerability to what?

Put coma or a preposition after “PILOTED SITES”

In the title and in other sections of the manuscript, the word “pattern” gives the impression that the study addressed vulnerability over time, while that is not the case. Thus, the word needs to be revised.

Abstract:

Line 23, the sentence “One of the hardly hit by the impacts are gender” must be revised.

1, the word hardly is the opposite of severely. 2, gender does not necessarily mean female/women.  

Line 24, genders’ vulnerability, I suggest women’s vulnerability instead

Line 26, the words in the bracket are not clear

Presentation of the results in the abstract is very poor and difficult to understand.

Key words: words that are already in the title are not required to be listed as key words.

Introduction:

Lines 46 to 47, why are “some social and vulnerable groups such as women” more exposed to climate change than other groups?

The authors described the argument presented on lines 60 to 62 as it is in line with their argument, while it is contrary to their argument. It requires revision.

Similarly, what the authors described their argument as a contrary to an argument presented on lines 70 to 72 is consistent with the argument to my understanding. This too requires revision.

Objective:

The objective is not clear. Is it to assess gendered vulnerability of household to climate change? to develop a new vulnerability index? Or both?

Study area:

Line 185, the word “respectively” is not needed.

Make the difference between the number of sites (3)? and the number of villages (6)? clear

In some cases, sites and villages are used interchangeably, e.g, lines 136 and 186

Be consistent with the abbreviation for Reducing Emissions from Deforestation and Degradation, REDD vs REDD+ and define it the first time it is mentioned/used.

Lines 189 to 190 what are the types of forests in sites Afi-Mbe and Ekuri-Iko? Is the name of the third site, Mangrove Forest?

Lines 203 to 204, REDD+ is described as a climate change adaptation measure. However, REDD+ as its name implies aims at mitigating climate change by reducing emissions of CO2. What component of REDD+ do the authors consider as climate change adaptation?

Study design:

Total number of respondents were 200 and these respondents are from 6 villages each stratified in to 3 strata totaling 18 number of categories/groups. This means that each category was represented by an average of 11 samples. How do the authors justify the representativeness of the samples? If this is not the case, the authors need to elaborate how number of samples per each category was decided.

Discussion and conclusion:

The implication of the results for REDD+ design is poorly discussed. The authors only indicated that it is useful but did not show/discuss how. In general, the relationship between the results of the study and REDD+ is poorly presented.

Reviewer 3 Report

The paper aims to identify the gender dimension of vulnerability, in the context of climate change, taking as study area the Cross River State, Nigeria. The topic is interesting, but the paper should undergo major revision, due to multiple motives - which I mentioned in the attached pdf. Also, before submitting the reviewed version, the authors should correct the English language and style errors. Currently, the manuscript is hard to read and poorly constructed, although it contains thought-provoking findings. Please find attached the pdf with the comments to be addressed. I hope they are useful and that they will positively contribute to the improvement of this research work.

Author Response

  1. Abstract
  2. Introduction
  3. IPCC Livelihood Vulnerability Index has been moved to sub-section in methodology
  4. Details about the population of the cluster in the study area, and on their climate conditions and climate change has been addressed in the study area.
  5. Map of the study site has been adjusted for more visibility.
  6. Details about REDD+ project has been added at beginning of the section.
  7. Formula for equation 1 has been adjusted as advised.
  8. Administration of research instrument and personnel engaged has been provided.
  9. LVI presentation has been moved from Introduction to sub-section in Methodology as advised.
  10. The Table has been changed to Landscape as advised and the complete has also been added at the appendix with survey questions.
  11. The result section has been re-written with more clarity
  12. Benefits and Limitations of the study has been added as advised
  13. Clarification of the terminology of Natural Disaster has been added at the bottom of the Table 2 according to https://www.undrr.org/terminology.
  14. Citation at the conclusion has been adjusted as advised.

Round 2

Reviewer 3 Report

I congratulate the authors for greatly improving their manuscript and appreciate their diligent manner of approaching this review. Only a few comments are to be addressed in this minor revision phase. Please find them attached. The main issue is the English language, which should be edited. I observed that the highlighted parts of the manuscript are very well written, so I ask the authors to read the entire paper again and make the corrections. Also, I advise the Editor to thoroughly check for proper English grammar and language style.

Author Response

Please find the attachment

This manuscript is a resubmission of an earlier submission. The following is a list of the peer review reports and author responses from that submission.